# TRANSFORMER-VQ: LINEAR-TIME TRANSFORMERS VIA VECTOR QUANTIZATION

**Lucas D. Lingle**
Independent Researcher
lucasdaxlingle@gmail.com

## ABSTRACT

We introduce Transformer-VQ, a decoder-only transformer computing softmax-based dense self-attention in linear time. Transformer-VQ's efficient attention is enabled by vector-quantized keys and a novel caching mechanism. In our large-scale experiments, Transformer-VQ is shown highly competitive in quality, obtaining 0.99 bpb on Enwik8, 26.6 ppl on PG-19, and 3.16 bpb on ImageNet64. In addition, the optimized implementation of Transformer-VQ is over 3x faster than a comparable quadratic-time transformer at sequence length 8k, is over 12x faster at 32k, and can scale to 131k with similar throughput. Code available: https://github.com/transformer-vq/transformer_vq

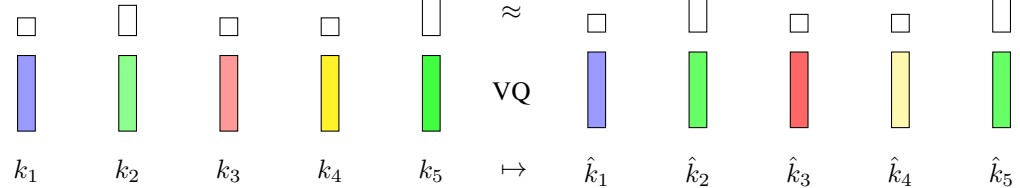

Figure 1: Schematic of the VQ-Attention approximation. The colorful and blank boxes depict the keys and attention weights, respectively. The keys on the right have been vector-quantized. Since the green keys $k_2, k_5$ map to the same code, they have the same attention weights in this attention head.

## 1 INTRODUCTION

Transformer (Vaswani et al., 2017) language models would ideally scale to long sequences, since their predictive abilities often improve as context length increases (Dai et al., 2019; Kaplan et al., 2020). Unfortunately, the standard transformer uses a self-attention mechanism with a quadratic time complexity with respect to sequence length. This limits the practicality of applying transformers to very long sequences, since increasing the sequence length by a factor of $10^n$ increases the attention computations by a factor of $100^n$. Transformer variants that overcome this efficiency bottleneck have the potential to facilitate new long-context applications and enable new breakthroughs.

Up to this point, a variety of *efficient transformers* (Tay et al., 2020b) have been proposed to scale to long sequences. Techniques include sparsity (Child et al., 2019; Ye et al., 2019; Beltagy et al., 2020; Kitaev et al., 2020; Qiu et al., 2020; Roy et al., 2021; Tay et al., 2020a; Sukhbaatar et al., 2021; Wu et al., 2022; Liu et al., 2023; Zhang et al., 2023), compression (Liu et al., 2018; Rae et al., 2020; Ainslie et al., 2020; Zhu et al., 2021; Ren et al., 2021; Nawrot et al., 2021; 2023), low-rank approximations (Wang et al., 2020; Vyas et al., 2020; Katharopoulos et al., 2020; Xiong et al., 2021; Tay et al., 2021; Choromanski et al., 2021), and cross-attention operations (Dai et al., 2019; Ma et al., 2021; Hutchins et al., 2022; Hawthorne et al., 2022). Other efficient sequence models have also been proposed (Gu et al., 2022; Lee-Thorp et al., 2022; Mehta et al., 2022; Smith et al., 2022; Hasani et al., 2022; Poli et al., 2023; Peng et al., 2023).

In this paper, we present Transformer-VQ, a transformer decoder with *dense self-attention computible in linear time* with respect to sequence length. This is made possible through a combination of vector-quantized keys, localized positional biases, and compressive cache that can be attended to efficiently, while yielding the same results as an uncompressed variable-length cache. Transformer-VQ is also simple to implement sampling for.

## 2 PRELIMINARIES

### 2.1 NOTATION

The real numbers are denoted by $\mathbb{R}$ and the extended real numbers $\mathbb{R} \cup \{-\infty, \infty\}$ by $\bar{\mathbb{R}}$. Zero-based indices are used for all tensors. When indexing a matrix $\mathbf{M}$ along the first axis, we use $\mathbf{M}_i$ to denote a column vector and $\mathbf{M}_{i,:}$ to denote a row vector. The functions $\text{LN}(\cdot), \text{Softmax}(\cdot), \text{Concat}(\cdot)$ denote LayerNorm (Ba et al., 2016), softmax, and concatenation, each applied row-wise. The symbols $\triangleq, \propto, \odot, \exp(\cdot), \delta_{a,b}, \text{SG}(\cdot)$ denote equality by definition, proportionality, element-wise product, element-wise exponentiation, Kronecker delta function, and the stop-gradient operator. We slightly abuse notation to write inner products of vectors $\mathbf{u}, \mathbf{v}$ as $\mathbf{u}^\top \mathbf{v}$, and outer products as $\mathbf{u}\mathbf{v}^\top$.

We assume familiarity with transformers (Vaswani et al., 2017), and use the notation $D_m$ to denote the model width, $D_k$ to denote the query/key vector width, and $D_v$ to denote the value vector width.

### 2.2 VECTOR QUANTIZATION

Vector quantization (VQ) is a technique used extensively throughout this work. In this subsection we briefly review vector quantization, motivate its use in self-attention, and discuss the backpropagation-compatible VQ scheme introduced by van den Oord et al. (2017).

### 2.3 VECTOR QUANTIZERS AND CODEBOOKS

**Definition 2.1.** A *vector quantizer* is a function $\text{VQ}(\cdot; \mathbf{C})$ with domain $\mathbb{R}^D$ and codomain $\mathbb{R}^D$. For an input $\mathbf{x}$, its output $\hat{\mathbf{x}}$ is given by

$$z \triangleq \arg\min_s ||\mathbf{x} - \mathbf{C}_s||^2 \tag{1}$$

$$\hat{\mathbf{x}} \triangleq \mathbf{C}_z \tag{2}$$

where $\mathbf{C} \in \mathbb{R}^{S \times D}$ is known as the *codebook*. The row indices $\{0, \dots, S-1\}$ of $\mathbf{C}$ are called *shortcodes*, and the rows themselves are called *codewords*.

**Theorem 2.2** (Based on Guo et al. (2019)). *Let $\mathbf{q} \in \mathbb{R}^D$ be a random variable with $\mathbb{E}_\mathbf{q}[\mathbf{q}\mathbf{q}^\top] = \sigma^2 \mathbf{I}_D$ for some $\sigma > 0$, and let $\mathbf{k} \in \mathbb{R}^D$ be a random variable independent of $\mathbf{q}$. Let $\varphi : \mathbb{R}^D \to \mathbb{R}^D$ be a deterministic function. Then*

$$\mathbb{E}_{\mathbf{q},\mathbf{k}}||\mathbf{q}^\top \mathbf{k} - \mathbf{q}^\top \varphi(\mathbf{k})||^2 \propto \mathbb{E}_\mathbf{k}||\mathbf{k} - \varphi(\mathbf{k})||^2. \tag{3}$$

**Corollary 2.3.** *Let the conditions of Theorem 2.2 hold. Given the constraint that $\varphi(\mathbb{R}^D) = \{\mathbf{C}_s\}_{s=0}^{S-1}$, the choice $\varphi(\cdot) = VQ(\cdot; \mathbf{C})$ minimizes $\mathbb{E}_{\mathbf{q},\mathbf{k}}||\mathbf{q}^\top \mathbf{k} - \mathbf{q}^\top \varphi(\mathbf{k})||^2$.*

**Corollary 2.4.** *Let the conditions of Theorem 2.2 hold. With $\hat{\mathbf{k}} = VQ(\mathbf{k}; \mathbf{C})$ we have*

$$\arg\min_\mathbf{C} \mathbb{E}_{\mathbf{q},\mathbf{k}}||\mathbf{q}^\top \mathbf{k} - \mathbf{q}^\top \hat{\mathbf{k}}||^2 = \arg\min_\mathbf{C} \mathbb{E}_\mathbf{k}||\mathbf{k} - \hat{\mathbf{k}}||^2. \tag{4}$$

*Remark* 2.5. Fnding the global minimizer $\mathbf{C}^* = \arg\min_\mathbf{C} \mathbb{E}_\mathbf{k}||\mathbf{k} - \hat{\mathbf{k}}||^2$ is expensive, so in practice we approximate it using the method from van den Oord et al. (2017); Razavi et al. (2019).

### 2.4 VECTOR-QUANTIZED REPRESENTATION LEARNING

**Definition 2.6** (Based on van den Oord et al. (2017)). A *vector-quantizer with straight-through estimator* is a function $\text{STVQ}(\cdot; \mathbf{C})$ with domain $\mathbb{R}^D$ and codomain $\mathbb{R}^D$. For an input $\mathbf{x}$, its output $\hat{\mathbf{x}}$ is given by

$$z \triangleq \arg\min_s ||\mathbf{x} - \mathbf{C}_s||^2 \tag{5}$$

$$\hat{\mathbf{x}} \triangleq \mathbf{x} + \text{SG}(\mathbf{C}_z - \mathbf{x}). \tag{6}$$

*Remark* 2.7. For any $\mathbf{x} \in \mathbb{R}^D$, $\text{STVQ}(\mathbf{x}; \mathbf{C})$ evaluates to $\text{VQ}(\mathbf{x}; \mathbf{C})$. However, for purposes of backpropagation, the Jacobian of the quantizer w.r.t. its input will now be an identity matrix everywhere, instead of a zero matrix almost everywhere. Intuitively, when using STVQ, gradients w.r.t. the quantizer outputs are copied 'straight through' to the inputs.

*Remark* 2.8. We overload the notation $\text{STVQ}(\cdot; \mathbf{C})$ to operate row-wise on matrix-valued inputs.

## 3 TRANSFORMER-VQ

We now propose Transformer-VQ, a decoder-only transformer that can compute dense self-attention in linear time. Proofs for all theoretical results are given in Appendix A.

### 3.1 QUADRATIC-TIME FORMULATION

**Definition 3.1.** *Vector-Quantized Self-Attention* is a function $\text{VQAttn}(\cdot; \mathbf{C}, \mathbf{W}_{\{Q,K,V,G,O\}})$ with domain $\mathbb{R}^{T \times D_m}$ and codomain $\mathbb{R}^{T \times D_m}$. For an input $\mathbf{X}$, its output $\mathbf{Y}$ is defined via

$$\tilde{\mathbf{X}} \triangleq \text{LN}(\mathbf{X}) \in \mathbb{R}^{T \times D_m} \tag{7}$$

$$\mathbf{Q} \triangleq \tau^{-0.5}\text{LN}(\tilde{\mathbf{X}}\mathbf{W}_Q) \in \mathbb{R}^{T \times D_k} \tag{8}$$

$$\mathbf{K} \triangleq \tau^{-0.5}\text{LN}(\tilde{\mathbf{X}}\mathbf{W}_K) \in \mathbb{R}^{T \times D_k} \tag{9}$$

$$\mathbf{V} \triangleq \phi_v(\tilde{\mathbf{X}}\mathbf{W}_V) \in \mathbb{R}^{T \times D_v} \tag{10}$$

$$\mathbf{G} \triangleq \phi_g(\tilde{\mathbf{X}}\mathbf{W}_G) \in \mathbb{R}^{T \times D_v} \tag{11}$$

$$\hat{\mathbf{K}} \triangleq \text{STVQ}(\mathbf{K}; \mathbf{C}) \in \mathbb{R}^{T \times D_k} \tag{12}$$

$$\mathbf{W} \triangleq \phi_w(\mathbf{Q}\hat{\mathbf{K}}^\top + \mathbf{B}) \in \mathbb{R}^{T \times T} \tag{13}$$

$$\mathbf{O} \triangleq (\mathbf{W}\mathbf{V}) \odot \mathbf{G} \in \mathbb{R}^{T \times D_v} \tag{14}$$

$$\mathbf{Y} \triangleq \mathbf{X} + \mathbf{O}\mathbf{W}_O \in \mathbb{R}^{T \times D_m} \tag{15}$$

where each $\mathbf{W}_\bullet$ denotes a trainable projection, $\mathbf{B}$ denotes positional biases and/or mask, $\tau$ is a fixed constant, and the $\phi_v, \phi_g, \phi_w$ are element-wise or row-wise nonlinearities. The query/key LayerNorms use unit gain and zero bias, and $\text{STVQ}(\cdot; \mathbf{C})$ denotes row-wise application of vector-quantization with a straight-through gradient estimator (van den Oord et al., 2017).

*Remark* 3.2. Our attention mechanism is applied to a gated attention unit (GAU) design inspired by Hua et al. (2022). GAU is a single-headed gated attention mechanism and generally uses a small key width $D_k = 128$, and a large value width $D_v = 2D_m$, with two GAUs replacing a single transformer layer. This yields a similar parameter count and compute requirement as the usual transformer layer.

*Remark* 3.3. Prior work has also applied LayerNorm or similar to the queries and keys in attention (Henry et al., 2020; Roy et al., 2021; Zhu et al., 2021; Wu et al., 2022; Hutchins et al., 2022; Dehghani et al., 2023; Elsen et al., 2023), generally finding it to improve numerical stability and convergence.

### 3.2 WARMUP: LINEAR-TIME ENCODER ATTENTION

To simplify the theorems for decoder-only attention and build intuition, we first discuss a setting where there is no causal mask.

**Theorem 3.4.** *Suppose* $\mathbf{B}_{i,j} = 0$ *for all* $i, j$, *and* $\phi_w$ *is an element-wise nonlinearity. Then the attention weights in Definition 3.1 can be factored:*

$$\mathbf{W} \triangleq \phi_w(\mathbf{Q}\hat{\mathbf{K}}^\top + \mathbf{B}) \tag{16}$$

$$= \phi_w(\mathbf{Q}\hat{\mathbf{K}}^\top) \tag{17}$$

$$= \phi_w(\mathbf{Q}\mathbf{C}^\top)\mathbf{\Delta} \tag{18}$$

*where* $\phi_w(\mathbf{Q}\mathbf{C}^\top) \in \mathbb{R}^{T \times S}$, $\mathbf{\Delta} \in \mathbb{R}^{S \times T}$ *and* $\mathbf{\Delta}_{s,t} \triangleq \delta_{s,z_t}$. *Here,* $\delta_{\cdot,\cdot}$ *denotes the Kronecker delta function and* $z_t$ *is the VQ shortcode for timestep* $t$.

**Theorem 3.5.** *Suppose* $\mathbf{B}_{i,j} = 0$ *for all* $i, j$, *and* $\phi_w$ *is the row-wise softmax nonlinearity. Then the attention weights in Definition 3.1 can be factored:*

$$\mathbf{W} \triangleq \phi_w(\mathbf{Q}\hat{\mathbf{K}}^\top + \mathbf{B}) \tag{19}$$

$$= \phi_w(\mathbf{Q}\hat{\mathbf{K}}^\top) \tag{20}$$

$$= \text{Diag}(\exp(\mathbf{Q}\mathbf{C}^\top)\mathbf{\Delta}\mathbf{1})^{-1}\exp(\mathbf{Q}\mathbf{C}^\top)\mathbf{\Delta} \tag{21}$$

*where* $\mathbf{1} \in \mathbb{R}^T$, $\text{Diag}(\exp(\mathbf{Q}\mathbf{C}^\top)\mathbf{\Delta}\mathbf{1})^{-1}\exp(\mathbf{Q}\mathbf{C}^\top) \in \mathbb{R}^{T \times S}$, $\mathbf{\Delta} \in \mathbb{R}^{S \times T}$ *and* $\mathbf{\Delta}_{s,t} \triangleq \delta_{s,z_t}$. *Here,* $\delta_{\cdot,\cdot}$ *denotes the Kronecker delta function and* $z_t$ *is the VQ shortcode for timestep* $t$.

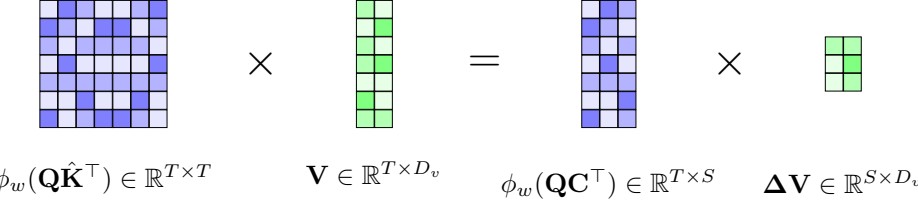

$$\phi_w(\mathbf{Q}\hat{\mathbf{K}}^\top) \in \mathbb{R}^{T \times T} \qquad \mathbf{V} \in \mathbb{R}^{T \times D_v} \qquad \phi_w(\mathbf{Q}\mathbf{C}^\top) \in \mathbb{R}^{T \times S} \qquad \mathbf{\Delta V} \in \mathbb{R}^{S \times D_v}$$

Figure 2: Schematic of the VQ-Attention factorization with element-wise $\phi_w$. The column set of $\mathbf{W} = \phi_w(\mathbf{Q}\hat{\mathbf{K}}^\top) \in \mathbb{R}^{T \times T}$ has size $\leq S$ due to VQ, so the attention output $\mathbf{O} = \mathbf{WV}$ can be obtained by computing the unique attention scores $\phi_w(\mathbf{Q}\mathbf{C}^\top)$ and using them to further aggregate to the grouped-sum $\mathbf{\Delta V}$. Transformer-VQ uses a softmax-based extension of this idea for its cache.

## 3.3 Linear-Time Decoder Attention

**Theorem 3.6.** *Let $L$ be a divisor of $T$. Suppose $\mathbf{B}_{i,j} = -\infty$ for $j > i$ (causal masking), and $\mathbf{B}_{i,j} = 0$ for $j < i - L$ (no bias outside a sliding window). Define $\mathbf{\Delta} \in \mathbb{R}^{S \times T}$ with $\mathbf{\Delta}_{s,t} \triangleq \delta_{s,z_t}$. Let $\phi_w$ be an element-wise nonlinearity with $\phi_w(-\infty) = 0$. For a tensor $\mathbf{M}$, let $\mathbf{M}_{(\dots,n,\dots)}$ denote the slice $\mathbf{M}_{\dots,nL:(n+1)L,\dots}$, where unsliced dimensions will be denoted by ':'. Then the product $\mathbf{WV}$ in Definition 3.1 can be computed using the following block-level recurrence:*

$$\mathbf{U}(n) \triangleq \begin{cases} \mathbf{U}(n-1) + \mathbf{\Delta}_{(:,n)}\mathbf{V}_{(n,:)} & \text{if } n \geq 0 \\ \mathbf{0} & \text{otherwise} \end{cases} \tag{22}$$

$$(\mathbf{WV})_{(n,:)} = \phi_w(\mathbf{Q}_{(n,:)}\mathbf{C}^\top)\mathbf{U}(n-2) \tag{23}$$

$$+ \phi_w(\mathbf{Q}^{(n,:)}\hat{\mathbf{K}}^\top_{(n-1,:)} + \mathbf{B}_{(n,n-1)})\mathbf{V}_{(n-1,:)} \tag{24}$$

$$+ \phi_w(\mathbf{Q}^{(n,:)}\hat{\mathbf{K}}^\top_{(n,:)} + \mathbf{B}_{(n,n)})\mathbf{V}_{(n,:)} \tag{25}$$

*where any tensor slice $\mathbf{M}_{(\dots,n,\dots)}$ is defined as a zero tensor of width $L$ in the sliced dimension if any block slice index $n$ is less than zero (zero-padding).*

**Theorem 3.7.** *Let the assumptions of Theorem 3.6 hold, but suppose $\phi_w$ is now the row-wise softmax nonlinearity. Let $\mathbf{1} \in \mathbb{R}^T$. Let $\mathbf{A} \triangleq \exp(\mathbf{Q}\hat{\mathbf{K}}^\top + \mathbf{B})$. Then the product $\mathbf{WV}$ in Definition 3.1 can be computed using the following block-level recurrence:*

$$\mathbf{U}(n) \triangleq \begin{cases} \mathbf{U}(n-1) + \mathbf{\Delta}_{(:,n)}\mathbf{V}_{(n,:)} & \text{if } n \geq 0 \\ \mathbf{0} & \text{otherwise} \end{cases} \tag{26}$$

$$\mathbf{L}(n) \triangleq \begin{cases} \mathbf{L}(n-1) + \mathbf{\Delta}_{(:,n)}\mathbf{1}_{(n)} & \text{if } n \geq 0 \\ \mathbf{0} & \text{otherwise} \end{cases} \tag{27}$$

$$(\mathbf{AV})_{(n,:)} = \exp(\mathbf{Q}_{(n,:)}\mathbf{C}^\top)\mathbf{U}(n-2) \tag{28}$$

$$+ \exp(\mathbf{Q}_{(n,:)}\hat{\mathbf{K}}^\top_{(n-1,:)} + \mathbf{B}_{(n,n-1)})\mathbf{V}_{(n-1,:)} \tag{29}$$

$$+ \exp(\mathbf{Q}_{(n,:)}\hat{\mathbf{K}}^\top_{(n,:)} + \mathbf{B}_{(n,n)})\mathbf{V}_{(n,:)} \tag{30}$$

$$(\mathbf{A1})_{(n)} = \exp(\mathbf{Q}_{(n,:)}\mathbf{C}^\top)\mathbf{L}(n-2) \tag{31}$$

$$+ \exp(\mathbf{Q}_{(n,:)}\hat{\mathbf{K}}^\top_{(n-1,:)} + \mathbf{B}_{(n,n-1)})\mathbf{1}_{(n-1)} \tag{32}$$

$$+ \exp(\mathbf{Q}_{(n,:)}\hat{\mathbf{K}}^\top_{(n,:)} + \mathbf{B}_{(n,n)})\mathbf{1}_{(n)} \tag{33}$$

$$(\mathbf{WV})_{(n,:)} = \text{Diag}((\mathbf{A1})_{(n)})^{-1}(\mathbf{AV})_{(n,:)}. \tag{34}$$

*Remark* 3.8. Theorem 3.7 provides an algorithm to compute VQ-Attention from the queries, keys, values, gates, and codebook in $\mathcal{O}(L(S + 2L)(D_k + D_v))$ time per query block, and therefore $\mathcal{O}(T(S + 2L)(D_k + D_v))$ time per sequence.

*Remark* 3.9. For numerical stability, we use an equivalent implementation of Theorem 3.7 that stores the running mean of the value vectors assigned to a given shortcode, instead of the sum as done by $\mathbf{U}(n-2)$. The result is made equivalent by moving the logarithm of the counts $\mathbf{L}(n-2)$ inside the exponentials $\exp(\mathbf{Q}_{(n,:)}\mathbf{C}^\top)$ appearing in $(\mathbf{AV})_{(n,:)}$ and $(\mathbf{A1})_{(n)}$. See pseudocode in Appendix E.

*Remark* 3.10. The general strategy of computing un-normalized softmax and its denominator is also used by many prior methods, including Memory-Efficient Attention (Rabe & Staats, 2021), FlashAttention (Dao et al., 2022), and RWKV (Peng et al., 2023); however, the first two techniques do not run in linear time, and the last one couples a recurrent state size to the model width, which is contrary to the principle of transformers.

### 3.4 Learning Algorithm

#### 3.4.1 Training Loss

Let $\boldsymbol{\theta}$ denote the set of non-codebook parameters of a transformer with $N$ VQ-Attention layers, and let $\mathcal{C} = \{\mathbf{C}^{(\ell)}\}_{\ell=0}^{N-1}$ denote the set of the layers' codebooks. For autoregressive modeling of a sequence $\mathbf{X} = \{\mathbf{x}_t\}_{t=0}^{T}$, we define the Transformer-VQ training loss as

$$\mathcal{L}(\mathbf{X}; \boldsymbol{\theta}, \mathcal{C}) = \mathcal{L}_{\mathrm{CE}}(\mathbf{X}; \boldsymbol{\theta}, \mathcal{C}) + \beta \mathcal{L}_{\mathrm{VQ}}(\mathbf{X}; \boldsymbol{\theta}, \mathcal{C}) \tag{35}$$

where $\beta > 0$ is a hyperparameter known as the commit loss coefficient, and

$$\mathcal{L}_{\mathrm{CE}}(\mathbf{X}; \boldsymbol{\theta}, \mathcal{C}) \triangleq \frac{1}{T} \sum_{t=0}^{T-1} -\ln p(\mathbf{x}_{t+1}|\mathbf{x}_{\leq t}, \boldsymbol{\theta}, \mathcal{C}) \tag{36}$$

$$\mathcal{L}_{\mathrm{VQ}}(\mathbf{X}; \boldsymbol{\theta}, \mathcal{C}) \triangleq \frac{1}{T} \sum_{t=0}^{T-1} \sum_{\ell=0}^{N-1} ||\mathbf{K}_t^{(\ell)} - \mathrm{SG}(\mathbf{C}_{z_t}^{(\ell)})||_2^2. \tag{37}$$

Thus, the training loss is the average next-token cross-entropy loss, plus the average token's commitment losses (van den Oord et al., 2017), summed over layer codebooks. The non-codebook parameters $\boldsymbol{\theta}$ receive a gradient from both loss terms. Following van den Oord et al. (2017); Razavi et al. (2019), codebooks are parameterized using EMA-smoothed k-means.

#### 3.4.2 Training Updates

Instead of updating on the full sequence loss given above, we generally update every $W/L$ query blocks, where $W \ll T$, which resembles a strategy used in prior works (Dai et al., 2019; Wu et al., 2022; Hutchins et al., 2022).

Each update is obtained by backpropagating through a window of $W$ timesteps, with gradients computed on the corresponding terms in the per-token average losses above. Codebooks are also updated every $W/L$ query blocks.

When $W/L = 1$, using Theorem 3.7 is an efficient equivalent to a variable-length key-value cache. When $W/L > 1$, a learning signal is sent through any value vectors added to the compressed cache within the backpropagation window.

## 4 Related Work

### 4.1 Hierarchical Attention

Combiner (Ren et al., 2021) proposes an approximation of softmax using a simple graphical model, and parameterizes its internal probabilities using max-pooling over query/key features, enabling decoder-only self-attention in subquadratic time. H-Transformer-1D (Zhu & Soricut, 2021) uses average-pooling operations over queries/keys to reduce the complexity of encoder-only self-attention. Transformer-LS (Zhu et al., 2021) uses dynamic projections to downsample long-range features in transformers by a user-specified factor. Hourglass Transformer (Nawrot et al., 2021) and MegaByte (Yu et al., 2023) eschew pooling in favor of convolutions or reshaping for temporal downsampling, and apply these techniques to reduce computation in the interior layers of decoder-only transformers.

Transformer-VQ differs from these works in that it uses vector quantization (VQ), a well-understood method for compression, instead of newly-designed heuristic methods. In addition, it does not rely on token contiguity to guide the compression process. Instead, it utilizes an equivalence to dense attention. Notably, Transformer-VQ is easier to sample from compared to previous hierarchical attention models; since the cache update logic can be equivalently applied every token instead of every $L$ tokens, there are no sporadic 'feature consolidation' operations required during sampling.

## 4.2 KERNELIZABLE ATTENTION

Kernelizable attention (Katharopoulos et al., 2020; Choromanski et al., 2021; Peng et al., 2021; Qin et al., 2022b) computes query and key features and applies the same nonlinearity to both of them separately, omitting additional nonlinearities when computing attention weights. By using the associativity of matrix multiplication, kernelized attention reduces attention to linear complexity. Transformer-VQ is distinguished from kernelizable attention through an asymmetric treatment of queries and keys, a deterministic equivalence to softmax-based attention, training stability, and strong quantitative results on long-context autoregressive modeling benchmarks.

Clustering attention (Vyas et al., 2020) uses vector-quantized queries and is also kernelizable. However, it requires learning per-layer codebooks for each sequence and uses a modified form of Lloyd's iterations based on Hamming distance and locality-sensitive hashing. This yields a complex non-causal algorithm which is only suitable for non-causal attention and is slow on TPUs. Transformer-VQ is strongly differentiated from clustering attention by its simplicity, applicability to decoder-only tasks, efficiency on TPUs, and large-scale experimental validation.

## 4.3 COMPRESSIVE ATTENTION

Compressive Transformers (Rae et al., 2020) directly learn a compression function for long-range features. LUNA (Ma et al., 2021) and Recurrent Transformers (Bulatov et al., 2022; Hutchins et al., 2022) use cross-attention to compress long-range features into a recurrent state. Notably, our model implements a kind of block-recurrent mechanism for its cache, but is significantly more parameter-efficient than the mechanisms proposed by Ma et al. (2021); Hutchins et al. (2022). More generally, Transformer-VQ differs from compressive/recurrent transformers in that it has an equivalence to quadratic-time attention over vector-quantized keys. In other words, if the keys are already vector-quantized, the Transformer-VQ cache losslessly reduces the cost to linear time.

Perceivers (Jaegle et al., 2021; Hawthorne et al., 2022) use cross-attention to attend to long sequences, and compute self-attention over only a narrow stack of 'latents'. Transformer-VQ differs from Perceivers in that it computes dense self-attention in linear time, instead of just cross-attention. Thus, while Perceivers' long-range layers incur a quadratic time complexity during sampling, Transformer-VQ generates sequences in linear time.

## 4.4 GATED SEQUENCE MODELS

Gated attention was introduced in FLASH (Hua et al., 2022) as a fusion of attention sublayers (Vaswani et al., 2017) and GLU-based MLP sublayers (Shazeer, 2020). Various gating mechanisms have previously been used to stabilize training of transformers (Parisotto et al., 2019) and other sequence models including S4 (Gu et al., 2022), GSS (Mehta et al., 2022), MEGA (Ma et al., 2023) and RWKV (Peng et al., 2023). Transformer-VQ uses the original gating formulation from Hua et al. (2022), and develops a new attention mechanism.

## 4.5 VQ, K-MEANS, AND BEYOND

Ideas relating to $k$-means, vector quantization, and/or codebooks have also been applied in transformers for sparse attention (Roy et al., 2021; Wang et al., 2021; 2022), feature learning (Mao et al., 2022; Roy et al., 2022), sparsely-activated MLPs (Lample et al., 2019), and expert selection (Roller et al., 2021). These works generally feature codebooks or similar *within* a transformer architecture. Several works also have proposed models that feature a codebook somewhere *outside* a transformer, e.g., when transformers are priors for VQ-VAEs (Kaiser et al., 2018; Dhariwal et al., 2020; Ramesh et al., 2021; Lee et al., 2022; Zhou et al., 2022). Transformer-VQ uses one codebook within each layer and, in contrast to all of the aforementioned works, computes dense self-attention in linear time.

Transformer-VQ is not directly related to methods which quantize the weights of a transformer e.g., Dettmers et al. (2022); Dettmers & Zettlemoyer (2023); Frantar et al. (2023). Such methods are typically applied after training to reduce the memory overhead of the model weights, while still computing in higher precision. As such, they do not affect the bitwidth of the queries, keys, or values, nor the complexity of self-attention. However, if applying our method to large models, these approaches may be complementary during inference.

## 5 EXPERIMENTS

Transformer-VQ is implemented in Jax (Bradbury et al., 2018) and Flax (Heek et al., 2023). For training, we use TPU v3 pod slices (Jouppi et al., 2017). Hyperparameters follow Appendix C unless specifically mentioned. Generated samples for all models are provided in Appendix D.

### 5.1 PRELIMINARY STUDIES

#### 5.1.1 CODEBOOK SIZE ABLATIONS

Larger codebook sizes may allow more flexible attention patterns and could improve the fidelity of the gradients, both of which are likely to benefit model quality at the expense of additional wall time. To investigate, we ablate the codebook size $S$ using the Enwik8 dataset (described in § 5.2.1), and report the lowest validation bits-per-byte (BPB, lower is better) obtained by each model in Table 1.

Table 1: Codebook size ablations.

| Setting | Val. BPB | Latency (Rel.) |
|---|---|---|
| $S = 256$ | 1.010 | **0.927** |
| $S = 512$ | 1.005 | 1.0 |
| $S = 1024$ | **1.000** | 1.109 |

Table 2: Compressive cache ablation.

| Compressive cache | Val. BPB | Latency (Rel.) |
|---|---|---|
| No | 1.026 | **0.836** |
| Yes | **1.010** | 0.927 |

Table 1 confirms the intuition that larger codebooks improve the prediction quality (lower BPB) in return for additional wall time per training step. In particular, for this dataset and model size, increasing the codebook size by a factor of two appears to decrease the validation BPB by about a factor of $0.995$. This result may suggest that the validation loss follows a power-law scaling (Kaplan et al., 2020) w.r.t. codebook size, though more experiments are needed to verify this phenomenon, and it is subject to the caveat that the validation loss must eventually level off (Henighan et al., 2020; Hoffmann et al., 2022), as the model cannot be expected to obtain zero loss at infinite codebook size.

#### 5.1.2 COMPRESSIVE CACHE ABLATION

Since our model has several architectural differences from most prior works, the benefit of the compressive cache must be shown directly. To investigate, we train a model with the compressive cache omitted, using codebook size $S = 256$. We report the validation BPB for Enwik8 in Table 2.

As shown in Table 2, removing the compressive cache reduces the wall time per step by a factor of about $1.1$ at the evaluated model size, but leads to a significant drop in quality (higher bits-per-byte). This confirms the importance of our compressive cache mechanism.

#### 5.1.3 LATENCY AND THROUGHPUT

We now measure the training latency (seconds per step) and compute the training throughput (tokens per second). The latter is computed as tokens per batch divided by latency, and allows a direct efficiency comparison across different sequence lengths. We benchmark on a TPU v3 with 8 cores, using a global batch size of 8 sequences. For these experiments, we scale the sequence length $T$ by multiples of $4\times$, and backpropagate through the entire sequence length.

We compare an unquantized quadratic-time full attention baseline ('Full') to our proposed linear-time VQ-Attention ('VQ') using Theorem 3.7. Since this theorem does not require access to the output gates, VQ-Attention can be extended to multi-head attention variants as well. For each attention type, we therefore benchmark three head types: multi-head attention ('MHA'; Vaswani et al. (2017)), multi-query attention ('MQA'; Shazeer (2019)), and single-head gated attention ('SHGA' aka GAU; Hua et al. (2022)). For VQ-attention, we use codebook size $S = 512$ and block length $L = 512$, which is the same as our later experiments. All models use roughly 190M parameters total.

As shown in Table 6, our model has a 3x lower latency/3x higher throughput than the quadratic attention baseline at $T = 8192$ when using SHGA for both. Moreover, Transformer-VQ is over 6x faster than the quadratic-time baselines when using MQA/MHA for both models. As the sequence

length increases to $T = 32768$, Transformer-VQ is over 12x faster than the quadratic time baseline when both use SHGA. For MQA/MHA, the quadratic-time attention gives an out-of-memory error at $T = 32768$, while Transformer-VQ maintains comparable or better throughput than with 4x shorter sequences. Table 7 even shows that Transformer-VQ can scale to sequences of length $T = 131072$ without a substantial decrease in throughput and without running out of memory.

## 5.2 QUANTITATIVE RESULTS

To assess the ability of Transformer-VQ to learn long-range dependencies, we now conduct a series of large-scale experiments, benchmarking on several long-range autoregressive modeling tasks. For fair comparison, we only benchmark against models (a) trained without using any extra data or augmentation, and (b) evaluated with fixed parameters. In all cases, we use codebook size $S = 512$.

### 5.2.1 ENWIK8

Enwik8 is a byte-level language modeling dataset consisting of 100 million bytes of unprocessed English-language Wikipedia articles (Mahoney, 2011), with long-term dependencies that may span tens of thousands of bytes. Per convention, it is split into train, validation, and test sets of 90 million, 5 million, and 5 million bytes, respectively (Child et al., 2019; Rae et al., 2020).

For this dataset, we trained a Transformer-VQ with 190M parameters, smaller than the model by Dai et al. (2019). We report test bits-per-byte (BPB) in Table 3.

Transformer-VQ obtains a BPB of 0.99, notably matching the result of the large Transformer-XL model from Dai et al. (2019), while using 33% fewer parameters and a 75% shorter cache that covers a longer context.

Table 3: Test bits-per-byte on Enwik8.

| Model | BPB |
|---|---|
| Dai et al. (2019) - XL | 0.99 |
| Child et al. (2019) - Sparse | 0.99 |
| Beltagy et al. (2020) - Longform. | 0.99 |
| Roy et al. (2021) - Routing | 0.99 |
| Sukhbaatar et al. (2019) - Adapt. | 0.98 |
| Nawrot et al. (2021) - Hourglass | 0.98 |
| Rae et al. (2020) - Compress. | 0.97 |
| Zhu et al. (2021) - Long-Short | 0.97 |
| Fan et al. (2020b) - Feedback | 0.96 |
| Lei (2021) - SRU++ | 0.95 |
| Sukhbaatar et al. (2021) - Expire. | 0.95 |
| Lutati et al. (2023) - Focus Attn. | **0.94** |
| Transformer-VQ | 0.99 |

For this dataset, we found overfitting was a significant issue, and due to the compressive cache mechanism, using i.i.d. attention dropout was not possible. Sweeping over the residual dropout rate, weight decay coefficient, and layerdrop (Fan et al., 2020a) rate, we found a setting yielding good generalization. Nonetheless Transformer-VQ does fall short of state-of-the-art here, with several works using complex recurrence or forgetting mechanisms and obtaining better Enwik8 results.

### 5.2.2 PG-19

PG-19 is an open-vocabulary language modeling dataset consisting of 11 gigabytes of text from over 28,000 freely-available Project Gutenberg books published prior to 1919 (Rae et al., 2020). The average number of words per book is nearly 70,000, enabling learning long-term dependencies, especially in novels (Sun et al., 2021; Hutchins et al., 2022).

For this dataset, we trained a Transformer-VQ with 1.3B parameters, similar to the largest model by Hutchins et al. (2022). Since PG-19 is an open-vocabulary dataset, we first learned a SentencePiece vocabulary (Kudo & Richardson, 2018) of size 32,000 using the BPE method. Following the calculations of Rae et al. (2020), we report the test set word-level perplexity (WLP) in Table 4.

Transformer-VQ obtains a WLP of 26.6, very close to the state-of-the-art by Block-Recurrent Transformers (Hutchins et al., 2022). Interestingly, since our Transformer-VQ design is equivalent to using dense self-attention with vector-quantized keys, our strong result shows that models using self-attention only (no recurrence) can also be highly competitive on PG-19. This affirms the efficacy of standalone self-attention as a method for sequence processing at scale. Furthermore, compared to the Block-Recurrent Transformer, our model can be implemented via intra-block sums and cross-block reductions, a strategy also used by FLASH (Hua et al., 2022) and shown to be faster in Appendix B. Lastly, we avoid the instabilities of FLASH (Qin et al., 2022a; Ma et al., 2023) thanks to softmax normalization and our cache normalization (§ 3.9).

Table 4: Test word-level perplexity on PG-19.

| Model | WLP |
| --- | --- |
| Yu et al. (2023) - MegaByte | 36.4 |
| Rae et al. (2020) - XL | 36.3 |
| Rae et al. (2020) - Compressive | 33.6 |
| Roy et al. (2021) - Routing | 33.2 |
| Hawthorne et al. (2022) - Perceiver AR | 28.9 |
| Hutchins et al. (2022) - Block-Recur. | **26.5** |
| Transformer-VQ | 26.6 |

Table 5: Validation bits-per-byte on ImageNet64.

| Model | BPB |
| --- | --- |
| Kingma et al. (2021) - VDM | 3.40 |
| Hawthorne et al. (2022) - Perceiver AR | 3.40 |
| Yu et al. (2023) - MegaByte | 3.40 |
| Grcic et al. (2021) - DenseFlow | 3.35 |
| Lipman et al. (2023) - Flow Matching | 3.31 |
| Hazami et al. (2022) - Efficient VDVAE | 3.30 |
| Transformer-VQ (190M) | **3.22** |
| Transformer-VQ (1.2B) | **3.16** |

### 5.2.3 IMAGENET64

ImageNet64 is an image dataset consisting of over 1.2 million images downsampled to 64x64 resolution (Chrabaszcz et al., 2017; Deng et al., 2009). Flattening the images yields an autoregressive density estimation task on sequences of over 12,000 bytes each. Note since the official test set is not public for this dataset, we report results on the official validation set. For validation purposes we used a held-out set of about 80,000 examples from the training split.

For this dataset, we trained Transformer-VQ models with 190M and 1.2B parameters, similar to the Enwik8 and PG-19 models, respectively. We report the bits-per-byte on the official validation set in Table 5. Several of the earlier baselines used an earlier variant of downsampled ImageNet prepared by van den Oord et al. (2016) with a different downsampling algorithm. Since that variant has been unavailable through official channels for about a year, we used the newer variant following Lipman et al. (2023). We emphasize that our results using the newer variant cannot be directly compared with baselines using the earlier variant; however, due to several reporting ambiguities, Table 5 does not symbolically distinguish variants used.

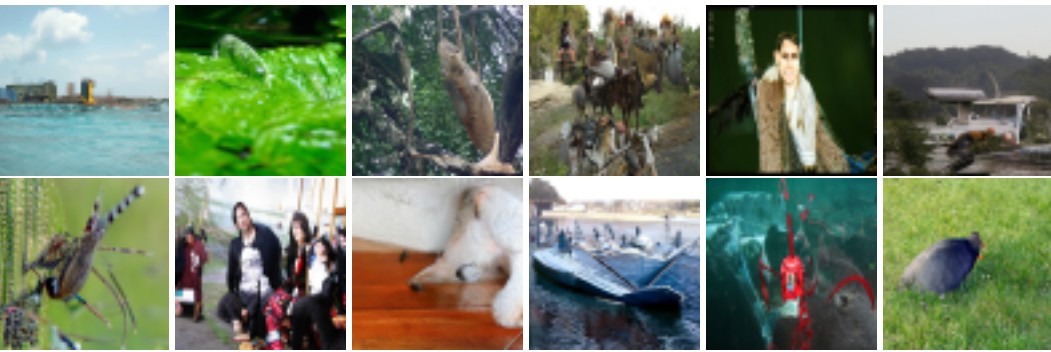

Figure 3: Generated samples from our large ImageNet64 model; nucleus 1.0.

Transformer-VQ with 190M parameters is roughly the same size as the Efficient VDVAE (Hazami et al., 2022), but obtains a better result of 3.22 BPB, setting a new state-of-the-art for small models. Transformer-VQ with 1.2B parameters obtains a 3.16 BPB, setting a new absolute state-of-the-art on this dataset, and generates high-fidelity samples on par with Perceiver AR while using 33% fewer steps, omitting its image-specific architectural adjustments, and generating samples in linear time.

## 6 CONCLUSION

Transformer-VQ is a transformer decoder computing softmax-based self-attention in linear time. Its efficient attention is enabled by vector-quantized keys, which allow our cache to be attended to in compressed form, while yielding the same result as uncompressed attention over the same keys. Large-scale experiments show Transformer-VQ is an efficient and flexible autoregressive model, with state-of-the-art results or near on PG-19 and ImageNet64. Future work directions include formal scaling laws, larger models, and porting to lower-level frameworks like Pallas, Triton, or CUDA.

REPRODUCIBILITY STATEMENT

To facilitate reproducibility, our attention mechanism is described mathematically in Section 3, and pseudocode is provided in Appendix E. In addition, our hyperparameters and other implementation details are given in Appendix C, and our implementation is open-sourced at the link in the abstract.

ACKNOWLEDGMENTS

We are grateful to the anonymous reviewers for their helpful feedback. In addition, we would like to express our gratitude to the Python community, especially the Jax ecosystem contributors, for the effective libraries used in this project. This project was generously supported with Cloud TPUs from Google's TPU Research Cloud (TRC).

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

## A    THEOREMS

### A.1    PROOF OF THEOREM 2.2

*Proof.* This proof is based on Guo et al. (2019). Invoking the fact that $\mathbf{q}, \mathbf{k}, \varphi(\mathbf{k}) \in \mathbb{R}^D$, the assumed independence between $\mathbf{q}$ and $\mathbf{k}$, the law of iterated expectations, and the isotropy assumption on $\mathbf{q}$, i.e., $\mathbb{E}_{\mathbf{q}}[\mathbf{q}\mathbf{q}^\top] = \sigma^2 \mathbf{I}_D$ for $\sigma^2 > 0$, we have

$$\mathbb{E}_{\mathbf{q},\mathbf{k}}[\mathbf{q}^\top \mathbf{k} - \mathbf{q}^\top \varphi(\mathbf{k})]^2 \tag{38}$$

$$= \mathbb{E}_{\mathbf{q},\mathbf{k}}[\mathbf{q}^\top (\mathbf{k} - \varphi(\mathbf{k}))]^2 \tag{39}$$

$$= \mathbb{E}_{\mathbf{q},\mathbf{k}}[\mathbf{q}^\top (\mathbf{k} - \varphi(\mathbf{k}))]^\top [\mathbf{q}^\top (\mathbf{k} - \varphi(\mathbf{k}))] \tag{40}$$

$$= \mathbb{E}_{\mathbf{q},\mathbf{k}}(\mathbf{k} - \varphi(\mathbf{k}))^\top \mathbf{q}\mathbf{q}^\top (\mathbf{k} - \varphi(\mathbf{k})) \tag{41}$$

$$= \mathbb{E}_{\mathbf{k}}(\mathbf{k} - \varphi(\mathbf{k}))^\top \mathbb{E}_{\mathbf{q}}[\mathbf{q}\mathbf{q}^\top](\mathbf{k} - \varphi(\mathbf{k})) \tag{42}$$

$$\propto \mathbb{E}_{\mathbf{k}}(\mathbf{k} - \varphi(\mathbf{k}))^\top \mathbf{I}_D(\mathbf{k} - \varphi(\mathbf{k})) \tag{43}$$

$$= \mathbb{E}_{\mathbf{k}}||\mathbf{k} - \varphi(\mathbf{k})||^2. \tag{44}$$

$\square$

### A.2    PROOF OF COROLLARY 2.3

*Proof.* By definition, $\mathrm{VQ}(\mathbf{k}; \mathbf{C}) \triangleq \arg\min_{\mathbf{c} \in \{\mathbf{C}_s\}_{s=0}^{S-1}} ||\mathbf{k} - \mathbf{c}||^2$. In other words, $\varphi(\mathbf{k}) = \mathrm{VQ}(\mathbf{k}; \mathbf{C})$ minimizes $||\mathbf{k} - \varphi(\mathbf{k})||^2$ under the constraint that the outputs of $\varphi$ are limited to the rows of $\mathbf{C}$, i.e., $\varphi(\mathbb{R}^D) = \{\mathbf{C}_s\}_{s=0}^{S-1}$. Since this choice is a pointwise minimizer under the given constraint, it is also a minimizer of the expectation $\mathbb{E}_{\mathbf{k}}||\mathbf{k} - \varphi(\mathbf{k})||^2$ under the same constraint.

Under the assumptions of Theorem 2.2, the aforementioned expectation is equal to $\mathbb{E}_{\mathbf{q},\mathbf{k}}||\mathbf{q}^\top \mathbf{k} - \mathbf{q}^\top \varphi(\mathbf{k})||^2$ up to a positive proportionality constant $\sigma^2$. As a result, $\mathrm{VQ}(\mathbf{k}; \mathbf{C})$ is also a minimizer of the expectation $\mathbb{E}_{\mathbf{q},\mathbf{k}}||\mathbf{q}^\top \mathbf{k} - \mathbf{q}^\top \varphi(\mathbf{k})||^2$ under the same constraint on the output of $\varphi$. $\square$

### A.3    PROOF OF THEOREM 3.4

*Proof.* When $\phi_w$ is an element-wise nonlinearity, $\phi(c)$ is well-defined, where $c$ is any scalar. Then using definitions alone, we have

$$[\phi_w(\mathbf{Q}\mathbf{C}^\top)\boldsymbol{\Delta}]_{i,j} = \phi_w(\mathbf{Q}\mathbf{C}^\top)_{i,:}\boldsymbol{\Delta}_{:,j} \tag{45}$$

$$= \sum_{s=0}^{S-1} \phi_w(\mathbf{Q}\mathbf{C}^\top)_{i,s}\boldsymbol{\Delta}_{s,j} \tag{46}$$

$$= \sum_{s=0}^{S-1} \phi_w(\mathbf{Q}_{i,:}\mathbf{C}_{s,:}^\top)\delta_{s,z_j} \tag{47}$$

$$= \phi_w(\mathbf{Q}_{i,:}\mathbf{C}_{z_j,:}^\top) \tag{48}$$

$$= \phi_w(\mathbf{Q}_{i,:}\hat{\mathbf{K}}_{j,:}^\top) \tag{49}$$

$$= [\phi_w(\mathbf{Q}\hat{\mathbf{K}}^\top)]_{i,j} \tag{50}$$

$\square$

### A.4    PROOF OF THEOREM 3.5

*Proof.* By Theorem 3.4 with $\phi_w(\cdot) = \exp(\cdot)$, we have $\exp(\mathbf{Q}\hat{\mathbf{K}}^\top) = \exp(\mathbf{Q}\mathbf{C}^\top)\boldsymbol{\Delta}$. Invoking the definition of row-wise softmax and applying substitution, we have

$$\mathrm{Softmax}(\mathbf{Q}\hat{\mathbf{K}}^\top) = \mathrm{Diag}(\exp(\mathbf{Q}\hat{\mathbf{K}}^\top)\mathbf{1})^{-1}\exp(\mathbf{Q}\hat{\mathbf{K}}^\top) \tag{51}$$

$$= \mathrm{Diag}(\exp(\mathbf{Q}\mathbf{C}^\top)\boldsymbol{\Delta}\mathbf{1})^{-1}\exp(\mathbf{Q}\mathbf{C}^\top)\boldsymbol{\Delta}. \tag{52}$$

$\square$

## A.5 Proof of Theorem 3.6

*Proof.* For $n = 0, 1$ the result follows by inspection.

For $n \geq 2$, by the definition of $\mathbf{U}(n-2)$ we have

$$\mathbf{U}(n-2) = \sum_{j=0}^{(n-1)L-1} \mathbf{\Delta}_{:,j} \mathbf{V}_{j,:} = \mathbf{\Delta}_{(:,0:n-1)} \mathbf{V}_{(0:n-1,:)}. \tag{53}$$

Note that in our notation, the superscripts' block index range is *non-inclusive* on the ending value, so $\mathbf{\Delta}_{(:,0:n-1)} \mathbf{V}_{(0:n-1,:)}$ is equal to the sum of the matrix products for the matrix blocks from 0 to $n-2$.

Thus, by substitution, we have

$$\phi_w(\mathbf{Q}_{(n,:)} \mathbf{C}^\top) \mathbf{U}(n-2) = \phi_w(\mathbf{Q}_{(n,:)} \mathbf{C}^\top) \mathbf{\Delta}_{(:,0:n-1)} \mathbf{V}_{(0:n-1,:)} \tag{54}$$

We invoke the same argument as in the proof of Theorem 3.4 to conclude $\phi_w(\mathbf{Q}_{(n,:)} \mathbf{C}^\top) \mathbf{\Delta}_{(:,0:n-1)} = \mathbf{W}_{(n,0:n-1)}$. Substituting this expression into the right-hand side above gives

$$\phi_w(\mathbf{Q}_{(n,:)} \mathbf{C}^\top) \mathbf{U}(n-2) = \mathbf{W}_{(n,0:n-1)} \mathbf{V}_{(0:n-1,:)}. \tag{55}$$

Substituting this expression into the formula for $(\mathbf{W}\mathbf{V})_{(n,:)}$ claimed in the theorem statement, and invoking the same argument as in the proof of Theorem 3.4 on the middle term, we see the claimed formula has the form $\mathbf{W}_{(n,0:n-1)} \mathbf{V}_{(0:n-1,:)} + \mathbf{W}_{(n,n-1)} \mathbf{V}_{(n-1,:)} + \mathbf{W}_{(n,n)} \mathbf{V}_{(n,:)}$. The diagonal block $\mathbf{W}_{(n,n)}$ of $\mathbf{W}$ is causally masked, so the sum of the three terms indeed equals $(\mathbf{W}\mathbf{V})_{(n,:)}$. □

## A.6 Proof of Theorem 3.7

*Proof.* Recall that we defined $\mathbf{A} \triangleq \exp(\mathbf{Q}\hat{\mathbf{K}}^\top + \mathbf{B})$. The proposed expression for $(\mathbf{A}\mathbf{V})_{(n,:)}$ follows from Theorem 3.6 with $\phi_w(\cdot) = \exp(\cdot)$. The proposed expression for $(\mathbf{A}\mathbf{1})_{(n)}$ follows by a substitution argument using $(\mathbf{A}\mathbf{V})_{(n,:)}$. Normalizing $(\mathbf{A}\mathbf{V})_{(n,:)}$ by $(\mathbf{A}\mathbf{1})_{(n)}$ and iterating over $n$ thus yields all blocks of the product $\mathbf{W}\mathbf{V}$ when the nonlinearity $\phi_w$ is row-wise softmax. □

# B  THROUGHPUT

We present throughput results for three methods to compute the cache variables: serial scan, matmul, and associative scan. The first two are generalizations of the cross-block reduction methods from FLASH (Hua et al., 2022), which were a simple cumulative sum and matrix multiplication by a lower-triangular matrix of ones, respectively. We found our proposed generalizations were necessary for stable training, an issue where FLASH has known weaknesses (Qin et al., 2022a; Ma et al., 2023). Pseudocode for each of our stable reduction methods is given in Appendix E.

In addition to the three cross-block reduction methods to compute the cache variables from parallel-computed per-block summaries, we also benchmark an *input scanning* implementation of VQ-Attention inspired by Wu et al. (2022); Hutchins et al. (2022), such that all the operations for a transformer layer are performed one input block at a time. To ground all comparisons, we benchmark the throughput against a transformer using unquantized quadratic-time attention, the same attention head type (SHGA, MQA, or MHA), and an identical non-codebook parameter count.

Table 6: Training throughput comparison (tokens/sec) on Google Cloud VM with 8 TPU v3 cores, between Full Attention and VQ-Attention with serial scan reduction.

| Model | Sequence Length | | | | | | | | | | | |
| | 2048 | | | 8192 | | | 32768 | | | 131072 | | |
| | Full | VQ | Speedup | Full | VQ | Speedup | Full | VQ | Speedup | Full | VQ | Speedup |
| SHGA | **65.5k** | 63.0k | 0.962× | 23.9k | **77.2k** | 3.230× | 6.5k | **82.1k** | 12.631× | OOM | OOM | – |
| MQA | 58.9k | **63.0k** | 1.070× | 10.4k | **74.8k** | 7.192× | OOM | **79.7k** | – | OOM | OOM | – |
| MHA | **52.0k** | 49.6k | 0.955× | 9.5k | **57.8k** | 6.084× | OOM | **60.7k** | – | OOM | OOM | – |

Table 7: Training throughput comparison (tokens/sec) on Google Cloud VM with 8 TPU v3 cores, between Full Attention and VQ-Attention with matmul reduction.

| Model | Sequence Length | | | | | | | | | | | |
| | 2048 | | | 8192 | | | 32768 | | | 131072 | | |
| | Full | VQ | Speedup | Full | VQ | Speedup | Full | VQ | Speedup | Full | VQ | Speedup |
| SHGA | **65.5k** | 62.5k | 0.954× | 23.9k | **75.2k** | 3.148× | 6.5k | **80.0k** | 12.250× | OOM | **69.5k** | – |
| MQA | 58.9k | **62.5k** | 1.061× | 10.4k | **74.9k** | 7.144× | OOM | **80.2k** | – | OOM | **67.7k** | – |
| MHA | **52.0k** | 48.2k | 0.926× | 9.5k | **58.2k** | 6.096× | OOM | **61.6k** | – | OOM | OOM | – |

Table 8: Training throughput comparison (tokens/sec) on Google Cloud VM with 8 TPU v3 cores, between Full Attention and VQ-Attention with associative scan reduction.

| Model | Sequence Length | | | | | | | | | | | |
| | 2048 | | | 8192 | | | 32768 | | | 131072 | | |
| | Full | VQ | Speedup | Full | VQ | Speedup | Full | VQ | Speedup | Full | VQ | Speedup |
| SHGA | **65.5k** | 58.9k | 0.899× | 23.9k | **62.2k** | 2.603× | 6.5k | **63.4k** | 9.754× | OOM | OOM | – |
| MQA | 58.9k | **62.8k** | 1.066× | 10.4k | **74.2k** | 7.134× | OOM | **79.0k** | – | OOM | **67.0k** | – |
| MHA | **52.0k** | 49.1k | 0.944× | 9.5k | **55.4k** | 5.831× | OOM | **57.8k** | – | OOM | OOM | – |

Table 9: Training throughput comparison (tokens/sec) on Google Cloud VM with 8 TPU v3 cores, between Full Attention and VQ-Attention, both with input scanning.

| Model | Sequence Length | | | | | | | | | | | |
| --- | --- | --- | --- | --- | --- | --- | --- | --- | --- | --- | --- | --- |
| | 2048 | | | 8192 | | | 32768 | | | 131072 | | |
| | Full | VQ | Speedup | Full | VQ | Speedup | Full | VQ | Speedup | Full | VQ | Speedup |
| SHGA | 32.0k | **40.8k** | 1.275× | 12.7k | **47.4k** | 3.732× | OOM | **49.4k** | – | OOM | OOM | – |
| MQA | 36.2k | **54.8k** | 1.514× | 14.5k | **64.9k** | 4.476× | OOM | **68.3k** | – | OOM | OOM | – |
| MHA | 30.0k | **43.7k** | 1.457× | 11.4k | **50.7k** | 4.447× | OOM | **53.1k** | – | OOM | OOM | – |

## C  TRAINING DETAILS

### C.1  HYPERPARAMETERS

Per-dataset hyperparameters are provided below.

Table 10: Hyperparameters.

| Name | Symbol | Enwik8 | PG-19 | ImageNet64 | ImageNet64 |
|---|---|---|---|---|---|
| parameter count | | 190M | 1.3B | 190M | 1.2B |
| global batch size | $B$ | 128 | 128 | 16 | 128 |
| sequence length | $T$ | 8192 | 8192 | 12288 | 12288 |
| backprop length | $W$ | 2048 | 2048 | 12288 | 2048 |
| block length | $L$ | 512 | 512 | 512 | 512 |
| model dimension | $D_m$ | 768 | 2048 | 768 | 2048 |
| key dimension | $D_k$ | 128 | 128 | 128 | 128 |
| value dimension | $D_v$ | 1536 | 4096 | 1536 | 4096 |
| num code | $S$ | 512 | 512 | 512 | 512 |
| num gau | $N$ | 48 | 48 | 48 | 48 |
| sinusoid dropout rate | $p_\text{dropsin}$ | 0.2 | 0.1 | 0.1 | 0.1 |
| residual dropout rate | $p_\text{dropres}$ | 0.5 | 0.1 | 0.0 | 0.0 |
| layerdrop rate | $p_\text{droplyr}$ | 0.3 | 0.1 | 0.0 | 0.0 |
| weight decay | | 0.0002 | 0.0 | 0.0 | 0.0 |
| optimizer | | adamw | adafactor | adamw | adafactor |
| total steps | | 125000 | 500000 | 125000 | 500000 |

Note that the 190M parameter ImageNet64 result was added after the other experiments had concluded. To avoid biasing its result, we use the exact same architectural hyperparameters as the Enwik8 model, and the exact same regularization as the larger ImageNet64 model. The smaller ImageNet model was trained in a newer version of our codebase optimized for higher throughput and faster compile times, rather than training on long sequences in constant space via input scans and truncated backprop through time. The attention in the optimized codebase was unit-tested to match the original.

### C.2  IMPLEMENTATION

Weights and token embeddings were initialized following Chowdhery et al. (2022). For the small model, the classifier layer omits LayerNorm and is independently parameterized. For the large model, the classifier layer uses LayerNorm and its projection is tied with the token embedding table, then scaled down by a large constant. For image datasets, we add absolute sinusoidal position embeddings, scaled by a trainable scalar, to the token embeddings (Hua et al., 2022; Vaswani et al., 2017). We used a maximum angular wavelength of $10^5$ for all sinusoidal embeddings.

We used the pre-norm placement of LayerNorm (Radford et al., 2019), and always used the RMS LayerNorm variant (Zhang & Sennrich, 2019). For the activations, we used $\phi_w =$ Softmax and $\phi_v = \phi_g =$ SiLU, the self-gated activation (Elfwing et al., 2017; Ramachandran et al., 2017). Several models use LayerDrop for regularization (Fan et al., 2020a), and following the Transformer-XL codebase (Dai et al., 2019) models apply dropout to the flipped sinusoidal embeddings used for (local) relative positional biases.

We used float32 parameters, with bfloat16 precision for most computations (Rae et al., 2021). For the AdamW optimizer (Loshchilov & Hutter, 2019), we used gradient clip 0.1, max learning rate $\alpha = 0.0004$ and hyperparameters $\beta_1 = 0.9, \beta_2 = 0.98, \epsilon = 10^{-9}$. For the Adafactor optimizer (Shazeer & Stern, 2018), we used relative stepsizes, update clip 1.0, max learning rate $\alpha = 0.01$, and hyperparameters $\hat{\beta}_1 = 0.0, \hat{\beta}_{2,t} = 1 - t^{-0.8}$. We used weight decay with a constant schedule throughout training and omit decay on any one-dimensional parameter tensors (Radford et al., 2019). The codebook commit coefficient was always $\beta = 0.0001$ and codebook EMA rate was always $\gamma = 0.99$. Learning rates were linearly warmed up for 10,000 steps, then decayed by a 10x factor using a cosine schedule.

## D    GENERATED SAMPLES

### D.1    QUALITATIVE ANALYSIS

#### D.1.1    PG-19

```
No effort has been made to explain elementary methods of photography, for the reason that such explanation has been
        found in the publications of every leading technical journal. The endeavor has been to present what is
        necessary to the amateur and the professional photographer, together with suggestions of how to make
        apparatus for the student, and to give a chapter on lens building. The author is fully aware of the
        imperfections in the methods described, and would like to emphasize the necessity of studying these methods
        carefully before attempting to use them, if it is desired to make satisfactory photographs. The most
        essential point in photography is the study of light. It is impossible to have success in photography unless
        the operator knows what light is. The writer believes that much may be done to advance the art of photography
         by the use of simple apparatus. The student must not overlook the fact that some simple apparatus is
        necessary in order to get good results. A lens is necessary to bring the image on the sensitive plate up to
        the focus of the lens. This lens is very expensive and only a few can be had of the best makers.
```

Figure 4: Sample excerpt from our PG-19 model, generated with nucleus 0.8.

We generated 128 sequences using nucleus sampling (Holtzman et al., 2020). In Figure 4, we observe a sample except in which our PG-19 model synthesizes high-quality text, and maintains a consistent tone, topic, and train of thought. These observations were found to hold for the vast majority of the samples we generated.

#### D.1.2    IMAGENET64

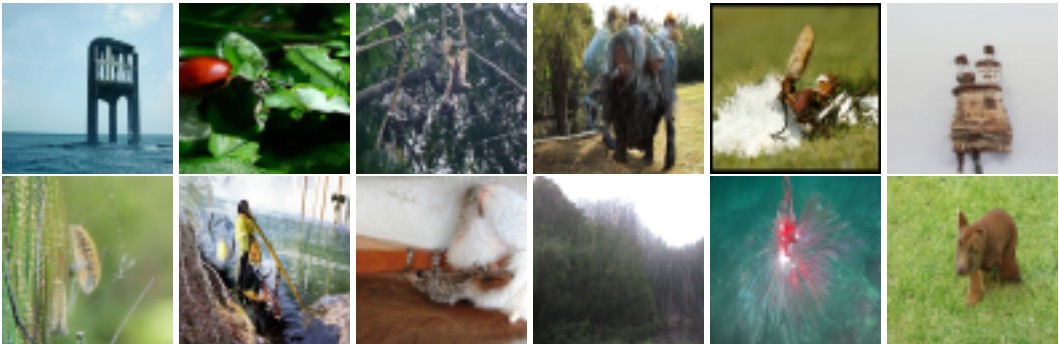

Figure 5: Generated samples from our large ImageNet64 model; nucleus 0.999.

Figures 3 and 5 show a subset of samples with the same indices from two batches with different nucleus settings. We see that our large ImageNet64 model synthesizes sequences of over 12,000 bytes and is capable of depicting relatively high-fidelity ocean water, shorelines, leaves, insects, trees, animals, people, mountains, and architecture.

### D.2    EXTENSIVE SAMPLES

Samples for Enwik8, PG-19, and ImageNet64 can be viewed at the anonymized URLs in Table 11.

Table 11: Generated samples' URLs by dataset.

| URL |
| --- |
| https://www.dropbox.com/sh/vu0dvw2bcglerwg/AADTQ9B4imAyEIc1Oo849v3ua?dl=0 |
| https://www.dropbox.com/sh/12civha5ulukulz/AAATnHL91RVax5kIb7QgS9ywa?dl=0 |
| https://www.dropbox.com/sh/xqr0q2e9seoz5wn/AADFnl1LWCaddC2CYRP3QSvpa?dl=0 |

### D.3    IMAGENET64 - FULL BATCH

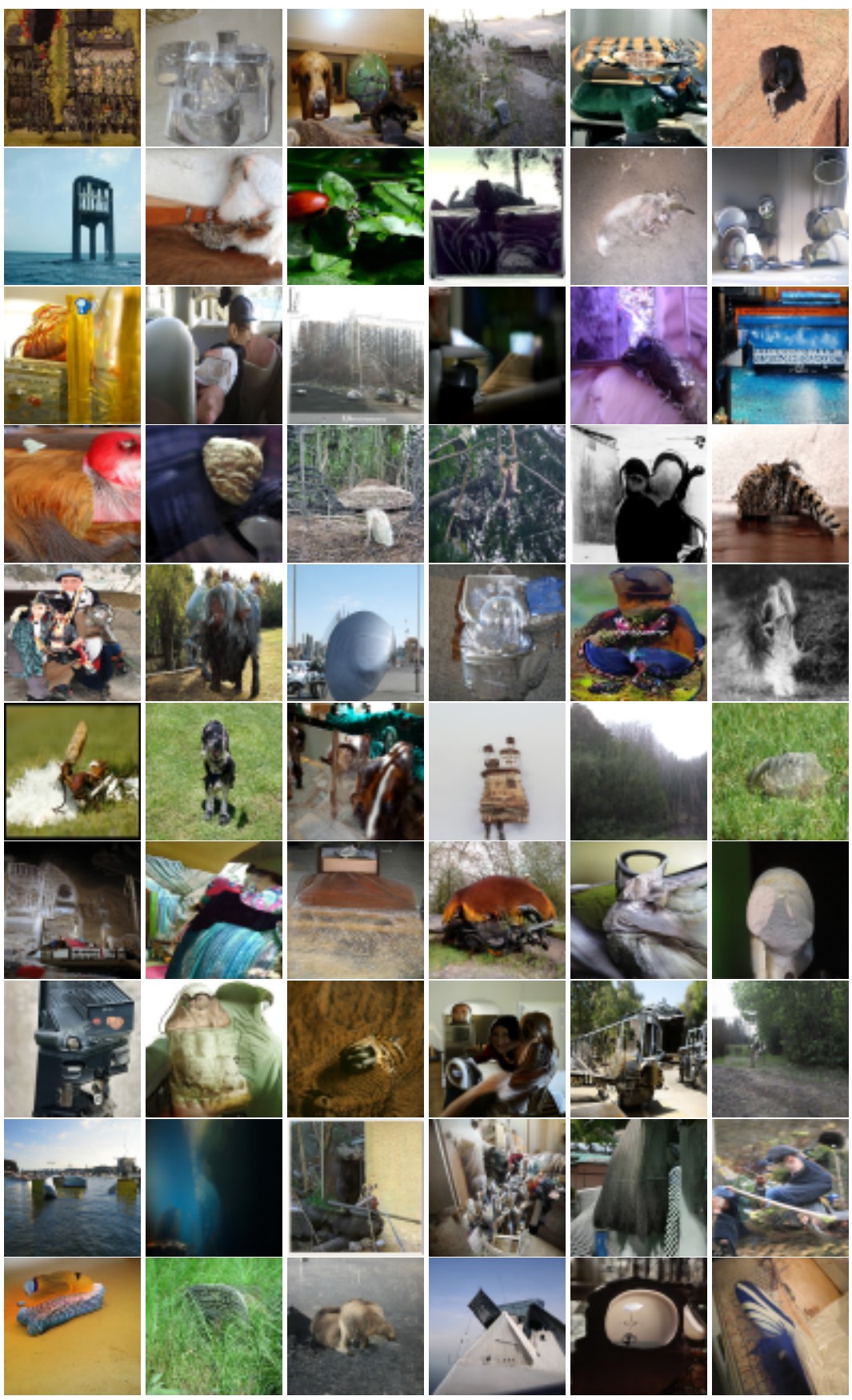

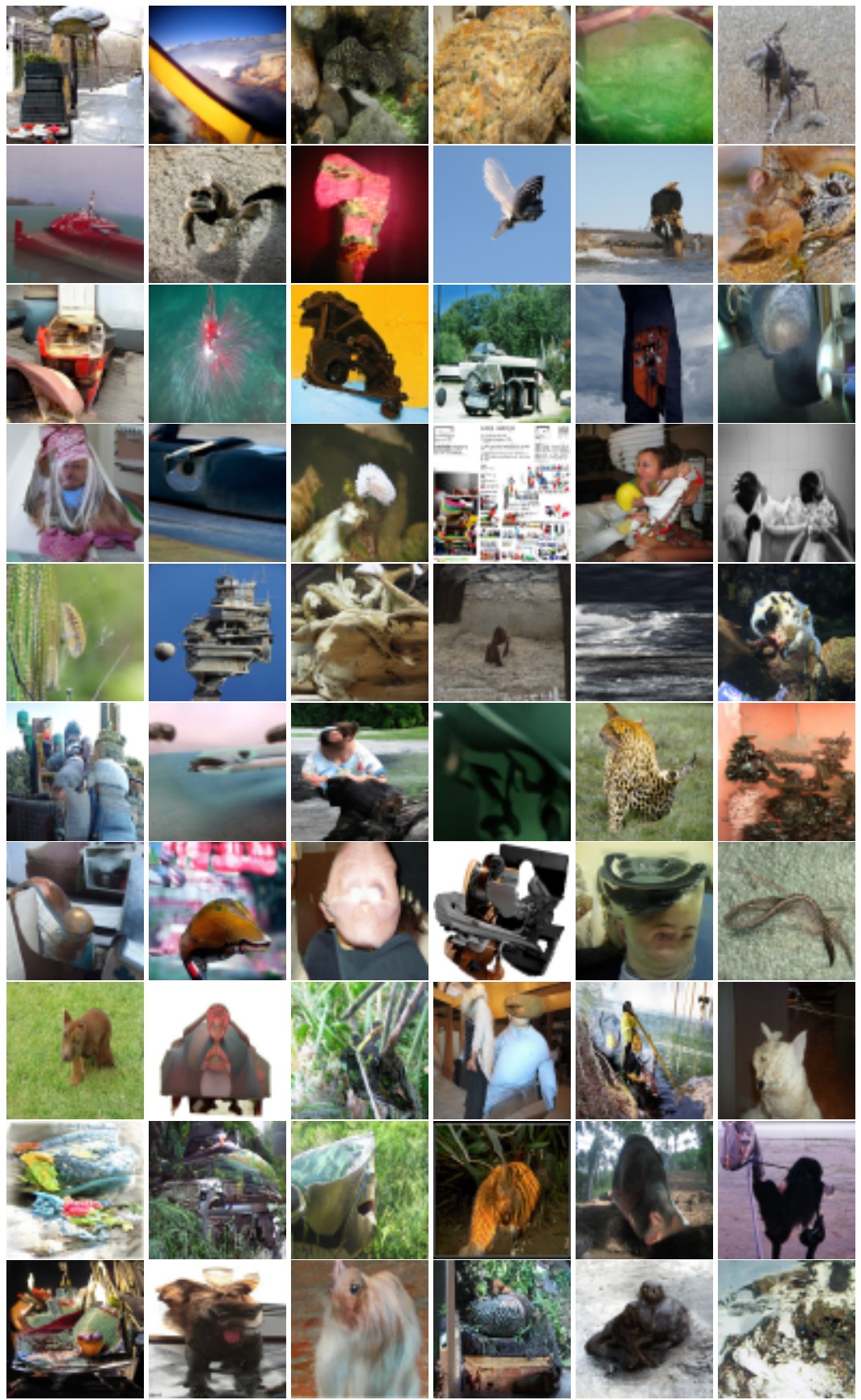

# E    PSEUDOCODE

```python
import flax.linen as nn
import jax
import jax.numpy as jnp
import chex

class VQAttn(nn.Module):
    n_code: int
    d_k: int
    d_v: int

    @nn.compact
    def __call__(self, x):
        """Input shape: [batch size, num blocks, block length, model width]."""
        B, R, C, D = x.shape
        S, K, V = self.n_code, self.d_k, self.d_v
        x_tilde = RMSLayerNorm(axis=-1)(x)
        q = RMSLayerNorm(axis=-1)(nn.Dense(self.d_k)(x_tilde))
        k = RMSLayerNorm(axis=-1)(nn.Dense(self.d_k)(x_tilde))
        v = jax.nn.silu(nn.Dense(self.d_v)(x_tilde))
        g = jax.nn.silu(nn.Dense(self.d_v)(x_tilde))
        quantizer = VectorQuantizer(codebook_size=self.n_code, width=self.d_k)
        k_hat, z, l_commit, l_codebook = quantizer(k) # quantized keys, shortcodes, etc
        c = quantizer.get_codebook()

        local_biases = XLBiasProducer(width=self.d_k, length=2*C)(q)
        chex.assert_shape(local_biases, [B, R, C, 2*C])
        local_biases_prev, local_biases_present = jnp.split(local_biases, 2, axis=-1)
        scores_present = jnp.einsum("brik,brjk->brij", q, k_hat)
        scores_present += local_biases_present
        scores_present -= 1e30 * (1 - jnp.tril(jnp.ones_like(scores_present)))

        k_hat_prev = jnp.pad(k_hat[:, :-1], ((0, 0), (1, 0), (0, 0), (0, 0)))
        v_prev = jnp.pad(v[:, :-1], ((0, 0), (1, 0), (0, 0), (0, 0)))
        scores_prev = jnp.einsum("brik,brjk->brij", q, k_hat_prev)
        scores_prev += local_biases_prev
        scores_prev = jnp.pad(
            scores_prev[:, 1:],
            ((0, 0), (1, 0), (0, 0), (0, 0)),
            constant_values=-1e30,
        )

        scores_cache = jnp.einsum("brik,sk->bris", q, c)
        cache_u_div_l_by_block, cache_l_by_block = get_cache_vars(z, v, S)
        chex.assert_shape(cache_u_div_l_by_block, [B, R, S, V])
        chex.assert_shape(cache_l_by_block, [B, R, S])
        count_biases = jnp.where(
            jnp.greater(cache_l_by_block, jnp.zeros_like(cache_l_by_block)),
            jnp.log(jnp.clip(cache_l_by_block, a_min=1.0)),
            jnp.full_like(cache_l_by_block, fill_values=-1e30),
        )
        scores_cache += jnp.expand_dims(count_biases, axis=-2)

        scores_present_max = jnp.max(scores_present, axis=-1)
        scores_prev_max = jnp.max(scores_present, axis=-1)
        scores_cache_max = jnp.max(scores_cache, axis=-1)
        scores_max = jnp.maximum(
            jnp.maximum(scores_present_max, scores_prev_max),
            scores_cache_max,
        )
        scores_max = jax.lax.stop_gradient(scores_max)
        scores_present -= scores_max[..., None]
        scores_prev -= scores_max[..., None]
        scores_cache -= scores_max[..., None]

        a_present = jnp.exp(scores_present)
        a_prev = jnp.exp(scores_prev)
        a_cache = jnp.exp(scores_cache)
        d = jnp.sum(a_present, axis=-1)
        d += jnp.sum(a_prev, axis=-1)
        d += jnp.sum(a_cache, axis=-1)
        w_present = a_present / d[..., None]
        w_prev = a_prev / d[..., None]
        w_cache = a_cache / d[..., None]
        wv = jnp.einsum("brij,brjv->briv", w_present, v)
        wv += jnp.einsum("brij,brjv->briv", w_prev, v_prev)
        wv += jnp.einsum("bris,brsv->briv", w_cache, cache_u_div_l_by_block)
        o = wv * g
        residual = nn.Dense(D)(o)
        return x + residual, l_commit, l_codebook
```

Code 1: Jax/Flax pseudocode for VQ-Attention.

```python
def get_cache_vars(z, v, n_code):
    # throughout this function, we often use clipping of elementwise denominators at 1 to avoid nans.
    # in the places where we do this, it does not alter the actual cache variable estimates
    # since the corresponding entries in the numerator will be zero when the clip is applied.

    delta = jax.nn.one_hot(z, num_classes=n_code, dtype=v.dtype, axis=-1)
    delta1_by_block = jnp.einsum("bris->brs", delta)
    deltav_by_block = jnp.einsum("bris,briv->brsv", delta, v)
    deltav_by_block_normalized = jnp.divide(
        deltav_by_block,
        jnp.clip(delta1_by_block[..., None], a_min=1.0),
    )

    def scan_func(carry, in_dict):
        # computes running average of the value vectors for each shortcode ("upper div lower"),
        # and running count ("lower").
        lower = carry["lower"]
        lower_block = in_dict["delta1_by_block"]
        lower_new = lower + lower_block
        f1 = jnp.divide(lower, jnp.clip(lower_new, a_min=1.0))
        f2 = jnp.divide(lower_block, jnp.clip(lower_new, a_min=1.0))
        upper_div_lower_new = jnp.add(
            f1[..., None] * carry["upper_div_lower"],
            f2[..., None] * in_dict["deltav_by_block_normalized"],
        )
        carry_new = dict(
            upper_div_lower=upper_div_lower_new,
            lower=lower_new,
        )
        return carry_new, carry_new  # state to carry, output to save

    # before we scan, we have to transpose since jax only supports scans along axis 0.
    # this is still fast, possibly because jnp.transpose might be choosing to return a view
    deltav_by_block_normalized = jnp.transpose(deltav_by_block_normalized, (1, 0, 2, 3))
    delta1_by_block = jnp.transpose(delta1_by_block, (1, 0, 2))
    _, cache_vars = jax.lax.scan(
        f=scan_func,
        init=dict(
            upper_div_lower=jnp.zeros(dtype=self.dtype, shape=deltav_by_block_normalized.shape[1:]),
            lower=jnp.zeros(dtype=self.dtype, shape=delta1_by_block.shape[1:]),
        ),
        xs=dict(
            deltav_by_block_normalized=deltav_by_block_normalized,
            delta1_by_block=delta1_by_block,
        ),
        unroll=1,
    )

    cache_var_upper_div_lower = jnp.pad(
        jnp.transpose(cache_vars["upper_div_lower"][:-2], (1, 0, 2, 3)),
        ((0, 0), (2, 0), (0, 0), (0, 0)),
    )
    cache_var_lower = jnp.pad(
        jnp.transpose(cache_vars["lower"][:-2], (1, 0, 2)),
        ((0, 0), (2, 0), (0, 0)),
    )
    return cache_var_upper_div_lower, cache_var_lower
```

Code 2: Jax/Flax pseudocode to get cache variables for all blocks; serial scan version.

```python
def get_cache_vars(z, v, n_code):
    # throughout this function, we often use clipping of elementwise denominators at 1 to avoid nans.
    # in the places where we do this, it does not alter the actual cache variable estimates
    # since the corresponding entries in the numerator will be zero when the clip is applied.

    delta = jax.nn.one_hot(z, num_classes=n_code, dtype=v.dtype, axis=-1)
    delta1_by_block = jnp.einsum("bris->brs", delta)
    deltav_by_block = jnp.einsum("bris,briv->brsv", delta, v)
    deltav_by_block_normalized = jnp.divide(
        deltav_by_block,
        jnp.clip(delta1_by_block[..., None], a_min=1.0),
    )

    delta1_by_block_tiled = jnp.einsum(
        "brs,bgs->bsrg",
        jnp.ones_like(delta1_by_block),
        delta1_by_block,
    )
    delta1_by_block_tiled = jnp.tril(delta1_by_block_tiled)
    delta1_fracs_by_block = jnp.divide(
        delta1_by_block_tiled,
        jnp.clip(jnp.einsum("bsrg->bsr", delta1_by_block_tiled)[..., None], a_min=1.0),
    )
    deltav_by_block_cumulative_normalized = jnp.einsum(
        "bsrg,bgsv->brsv", delta1_fracs_by_block, deltav_by_block_normalized
    )
    delta1_by_block_cumulative = jnp.cumsum(delta1_by_block, axis=1)

    cache_var_upper_div_lower = jnp.pad(
        deltav_by_block_cumulative_normalized[:, :-2],
        ((0, 0), (2, 0), (0, 0), (0, 0)),
    )
    cache_var_lower = jnp.pad(
        delta1_by_block_cumulative[:, :-2], ((0, 0), (2, 0), (0, 0))
    )
    return cache_var_upper_div_lower, cache_var_lower
```

Code 3: Jax/Flax pseudocode to get cache variables for all blocks; matmul version

```python
def get_cache_vars(z, v, n_code):
    # throughout this function, we often use clipping of elementwise denominators at 1 to avoid nans.
    # in the places where we do this, it does not alter the actual cache variable estimates
    # since the corresponding entries in the numerator will be zero when the clip is applied.

    delta = jax.nn.one_hot(z, num_classes=n_code, dtype=v.dtype, axis=-1)
    delta1_by_block = jnp.einsum("bris->brs", delta)
    deltav_by_block = jnp.einsum("bris,briv->brsv", delta, v)
    deltav_by_block_normalized = jnp.divide(
        deltav_by_block,
        jnp.clip(delta1_by_block[..., None], a_min=1.0),
    )

    def merge_func(a, b):
        a_upper_div_lower = a[0]
        b_upper_div_lower = b[0]
        a_lower = a[1]
        b_lower = b[1]
        lower_new = a_lower + b_lower
        term1 = jnp.multiply(
            jnp.divide(a_lower, jnp.clip(lower_new, a_min=1.0))[..., None],
            a_upper_div_lower,
        )
        term2 = jnp.multiply(
            jnp.divide(b_lower, jnp.clip(lower_new, a_min=1.0))[..., None],
            b_upper_div_lower,
        )
        upper_div_lower_new = term1 + term2
        return upper_div_lower_new, lower_new

    assoc_scan_output = jax.lax.associative_scan(
        fn=merge_func,
        elems=(deltav_by_block_normalized, delta1_by_block),
        reverse=False,
        axis=1,
    )
    deltav_by_block_normalized_cumulative = assoc_scan_output[0]
    delta1_by_block_cumulative = assoc_scan_output[1]
    cache_var_upper_div_lower = jnp.pad(
        deltav_by_block_normalized_cumulative[:, :-2],
        ((0, 0), (2, 0), (0, 0), (0, 0)),
    )
    cache_var_lower = jnp.pad(
        delta1_by_block_cumulative[:, :-2], ((0, 0), (2, 0), (0, 0))
    )
    return cache_var_upper_div_lower, cache_var_lower
```

Code 4: Jax/Flax pseudocode to get cache variables for all blocks; associative scan version

