# OpenReview forum: "Transformer-VQ: Linear-Time Transformers via Vector Quantization"
_ICLR.cc/2024/Conference — ICLR 2024 poster_

### Official Review · Reviewer_gEU7 · 2023-10-27

**Soundness:** 3 good
**Presentation:** 3 good
**Contribution:** 3 good
**Rating:** 8
**Confidence:** 3

**Summary:**

The paper proposes a novel attention mechanism with linear time complexity. The core idea is to apply vector quantization (VQ) on Keys, leading to the primary computing being (Query x Codebooks), where the number of codebooks is relatively small (256, 512, 1024) compared to the entire sequence length. The proposed method, termed Transformer-VQ, is applicable for both Transformer Encoder and Decoder models. Experimental results in long-context language modeling and image density estimation show that the proposed attention modification achieves performance comparable to prior works.

**Strengths:**

* While the idea of adopting VQ into Transformers is not entirely new (ex: Clustered Attention), the paper’s innovation is in using VQ for Keys and demonstrating its effectiveness (especially for Decoder). This is a nice contribution.
* The paper provides a comprehensive review of related works, addressing various aspects. Furthermore, the equations and notations are presented clearly to help clarify the core idea.

**Weaknesses:**

* Although this paper focuses on efficient computation, there is no direct comparison of FLOPs or actual inference latency on GPUs/TPUs with previous works. It would also be beneficial to include a comparison between models with and without VQ.
* Only one model size (1.3B for PG-19, 1.2B for ImageNet64) is used for experiments. To demonstrate the effectiveness, it would be valuable to test with multiple (smaller) model sizes where the sequence length exceeds the hidden dimension of Transformers. I assume that the actual inference speedup would arise when the sequence length is sufficiently larger than the hidden dimension of Transformers – I’m not sure if the tested models are such a case.

**Questions:**

* As the authors mentioned in Section 5.2.1, it seems that simple attention dropout is not applicable; is searching for the appropriate regularization setting a challenging job? In other words, is the model robust enough regarding regularization hyperparameters?
* The paper uses the GAU Transformer instead of a vanilla Transformer; Is there any reason why you used this variant? If the reason is that GAU employs single-headed attention, is Transformer-VQ applicable to a base (naïve) Transformer architecture?
* Linear-time Encoder is also explained in Section 3.2 but there are no corresponding experiments.
* The “truncation-free fixed-size cache” part is confusing. Does this refer to the limitation of the total stored Key size to “LK” (Section 3.4.2)? If so, then where does the ‘truncation-free’ part come?
* Suggestion: It would be beneficial to include a figure of the computation process (especially visualizing L, C, K, and V) for both the encoder and decoder. The current version only presents mathematical formulations, which can make it difficult to understand the core algorithm.
* (minor) In Section 3.1, maybe the “gated activation unit” is a typo of “gated attention unit”?
* (minor) Large “K” seems to be used in two different contexts (key and query block size); consider changing the latter to another letter.

---

> ### Author Response · Authors · 2023-11-16
> **Reply to reviewer gEU7**
>
> Thank you for your comments. We have recently uploaded a new version of our paper, which we believe is greatly improved; see our comment at the top for a summary of changes. We will now address your individual comments.
>
> >  The proposed method, termed Transformer-VQ, is applicable for both Transformer Encoder and Decoder models.
>
> To clarify, our proposal is a transformer decoder, similar to in spirit to Transformer-XL [1] or Memorizing Transformer [2], but whose cache contains the appropriate information while also being fixed in size. We do not make any claims about the efficacy of the encoder-only attention mentioned in the warmup; it is introduced only to build intuition for, and simplify, the proofs for decoder-only attention. We have added a brief comment in the encoder-only warmup subsection to make this clearer.
>
> > Although this paper focuses on efficient computation, there is no direct comparison of FLOPs or actual inference latency on GPUs/TPUs with previous works. It would also be beneficial to include a comparison between models with and without VQ.
>
> We have updated the paper to include a benchmark of the latency and throughput on a TPU v3 with 8 cores. Our model trains 3x faster at length 8192 than the baseline quadratic-time transformer, and 12x faster at length 32768. This is roughly in line with the ratio T^{2}/TS, since a 4x increase in sequence length led to a 4x increase in compute time.
>
> In addition, our model scales to a sequence length of 131072 without running out of memory and while maintaining comparable throughput.
>
> We also sought to include an end-to-end FLOP/s estimate for our model's train_op, or better yet an estimate of the MFU [3]. Unfortunately, Jax's analysis tool for FLOP count [4] is still a work in progress, and it does not return any result on TPUs. On CPUs, the backend does succeed in running an analysis, and the FLOP count is about 2x lower for our model than the baseline at length $T = 8192$. However, the outputted FLOP count estimates are also far below the product of the batch size in tokens and the model's non-embedding/non-codebook parameter counts. We unit tested using the same input batch, and verified all the non-embedding, non-codebook parameters changed after the training step. Since the gradients of the per-token losses must be summed, this shows that that the FLOP count estimate is wrong. As a result, we have not yet included the FLOP/s or MFU estimate in our paper yet. However, we will endeavor to raise this issue with the Jax developers and will update the paper if it is solved in time.
>
> > Only one model size (1.3B for PG-19, 1.2B for ImageNet64) is used for experiments.
>
> Actually, that's not true. In our Enwik8 experiment, we noted that our model size is 190M parameters.
>
> > it would be valuable to test with multiple (smaller) model sizes where the sequence length exceeds the hidden dimension of Transformers. I assume that the actual inference speedup would arise when the sequence length is sufficiently larger than the hidden dimension of Transformers – I’m not sure if the tested models are such a case.
>
> The sequence length always exceeds the hidden dimension. We assume you're referring to the heuristics like $6D_{m}$ or $12D_{m}$ minimum until attention compute "matters" at all. For short sequences of length $T=2048$, the slowdown is negligible, and for  $T=8192$ there is already a 3x speedup, as shown Table 6 of our updated paper. Note that for the $6D_{m}$ heuristic, we have $6D_{m} = 6 * 768 = 4608$ for our 190M parameter model.
>
> > As the authors mentioned in Section 5.2.1, it seems that simple attention dropout is not applicable; is searching for the appropriate regularization setting a challenging job? In other words, is the model robust enough regarding regularization hyperparameters?
>
> Right: if you apply attention dropout, it will elide all the cached timesteps whose keys land in a particular voronoi cell, and in practice that didn't work well. We found that residual dropout and LayerDrop [5] as a pair with drop rates of about (r, r/2) seem to be an effective alternative to attention dropout and MLP fan-out dropout as a pair with drop rates (r, r).
>
> Searching for optimal regularization is always a nuisance, but it is not particularly more so with our model. One helpful caveat however is that the weight decay setting should be set lower than usual, since the model already happens to have a tiny bit of regularization from the commitment loss.
>
> Due to space constraints, we will address the remainder of your questions in a separate post.
>
> References:
>
> [1] https://arxiv.org/abs/1901.02860
>
> [2] https://arxiv.org/abs/2203.08913
>
> [3] https://arxiv.org/abs/2204.02311
>
> [4] https://jax.readthedocs.io/en/latest/aot.html
>
> [5] https://arxiv.org/abs/1909.11556

---

> > ### Author Response · Authors · 2023-11-16
> > **Update: added a diagram**
> >
> > As per our promise, we have updated the paper with a schematic illustration of the VQ-attention approximation, and captioned it in a way that explains VQ-Attention intuitively.

---

> ### Author Response · Authors · 2023-11-16
> **Reply to reviewer gEU7, continued.**
>
> > The paper uses the GAU Transformer instead of a vanilla Transformer; Is there any reason why you used this variant?
>
> We used GAU because it is faster than multi-head attention (MHA) and yielded slightly better preliminary results than multi-query attention (MQA, [6]).
>
> > If the reason is that GAU employs single-headed attention, is Transformer-VQ applicable to a base (naïve) Transformer architecture?
>
> Yes it is. Theorem 3.7 only requires queries, quantized keys, shortcodes, values, and a codebook, so it can be applied with MHA or MQA instead of GAU.
>
> For stability we do recommend using layer-normalized keys, but this has been done in recent works such as Persimmon 8B [7] and to our knowledge has a neutral or positive impact on model quality in general.
>
> > Linear-time Encoder is also explained in Section 3.2 but there are no corresponding experiments.
>
> As we clarified above, it was introduced to build intuition and simplify the proofs later. Our paper proposes a new decoder-only model.
> If you want to apply it to encoder-only tasks, sort of like S4 does for their autoregressive model [8], you can, but we have not tried this.
>
> > The “truncation-free fixed-size cache” part is confusing. Does this refer to the limitation of the total stored Key size to “LK” (Section 3.4.2)? If so, then where does the ‘truncation-free’ part come?
>
> It refers to the fact that the cache variables $U(n-2)$ and $L(n-2)$ are of fixed size, yet together with key block $n-1$ and key block $n$, they contain all the information required to compute quantized-key full attention for query block $n$.
>
> Truncation-free is a reference to prior works, such as Transformer-XL, which implement a sliding window cache that effectively "truncates" the attention range in any given layer, and ultimately, for the entire model.
>
> > Suggestion: It would be beneficial to include a figure of the computation process (especially visualizing L, C, K, and V) for both the encoder and decoder. The current version only presents mathematical formulations, which can make it difficult to understand the core algorithm.
>
> We hope to have a satisfactory diagram soon, and will comment to notify you when we have included one.
>
> > (minor) In Section 3.1, maybe the “gated activation unit” is a typo of “gated attention unit”?
>
> > (minor) Large “K” seems to be used in two different contexts (key and query block size); consider changing the latter to another letter.
>
> Fixed both of these, thanks.
>
> References:
>
> [6] https://arxiv.org/abs/1911.02150
>
> [7] https://www.adept.ai/blog/persimmon-8b
>
> [8] https://arxiv.org/abs/2111.00396

---

> ### Author Response · Authors · 2023-11-17
> **Update on flop count issue**
>
> We are writing to update you that we have identified the source of error in the flop count estimate and have opened a ticket anonymously for the Flax developers to address. You can follow it here: https://github.com/google/flax/issues/3492

---

> ### Author Response · Authors · 2023-11-17
> **Updates on efficiency benchmarks, pseudocode, proof clarity**
>
> Since you had mentioned adding efficiency benchmarks in your review, we are writing to let you know we have added a number of throughput comparisons for various methods of computing the cache variables, and we benchmark against quadratic-time attention, using GAU/MQA/MHA for each of them.
>
> In these efficiency benchmarks, we observed that our model was over 3x faster than quadratic-time attention at length 8192, over 12x faster at length 32768, and could even scale to length 131072 with comparable throughput. Furthermore, we find that the empirically fastest method for computing cache variables also happens to be the one that runs in linear time. Finally, we also showed that input scanning by the attention layer--similar to what is done by Memorizing Transformer and Block-Recurrent Transformers--tends to be a bit slower.
>
> Since you mentioned evaluating on smaller models, we wanted to let you know that in addition to using a small model for Enwik8, we also added results for a small ImageNet64 model, which does quite well, and sets a new SoTA 3.22 BPB among small models.
>
> Since you mentioned the core algorithm being a bit difficult to understand from the mathematical formalisms, we wanted to let you jnow we have cleaned up the proofs as requested by reviewer 1jb5. In addition, we added pseudocode for all methods of computing the cache variables, as well as for VQ Attention itself, and we have clarified some ambiguities in the remarks after Theorem 3.7 about how we keep everything numerically stable.
>
> While we have a diagram that illustrates the intuitive idea of VQ-Attention, we have yet to create a diagram showing precisely how the cache interacts with everything else. We are working on this now, and will comment to let you know when we have included it.
>
> In the meantime, if there's anything else we can do to help, please let us know!

---

> ### Comment · Reviewer_gEU7 · 2023-11-18
> **Thank you for the effort**
>
> First, the authors addressed most of my concerns and corrected my misunderstandings. The revised version shows an improvement in readability and clarifies the contributions. (I still think Figure 1 is not the best way to illustrate the key idea, but I believe it can be also improved).
>
> Second, the authors added lots of experiments, particularly those related to actual efficiency. I appreciate the extensive effort in conducting such experiments, as they contribute significantly to the importance of this paper.
>
> Based on the substantial improvements made in response to my review, I'm increasing my score from 6 to 8.

---

> ### Author Response · Authors · 2023-11-20
> **Thank you for review**
>
> Thank you for the increase!
>
> We have yet to come up with a satisfactory second diagram, but are hoping to have something by the end of the day.
>
> The main issue is that there are a lot of relevant pieces. We are considering just using a diagram similar Figure 1 of the Performer paper [1], which shows a factorization of the unnormalized attention result, in our case we essentially have $\exp(QK^T)V = \exp(QC^T)\Delta V$, ignoring issues such as the causal mask, etc. Do you feel that would be sufficient?
>
> References:
>
> [1] https://arxiv.org/abs/2009.14794

---

> > ### Author Response · Authors · 2023-11-21
> > **Added additional diagram**
> >
> > We have added an additional diagram, as described above. It shows the attention output can be factored as $\exp(Q\hat{K}^T)V = \exp(QC^T)\Delta V$ in the absence of causal masking or positional biases. This observation generalizes directly to our cache mechanism. Thanks for your feedback, and we hope you find the new diagram satisfactory!

---

### Official Review · Reviewer_1jb5 · 2023-10-29

**Soundness:** 4 excellent
**Presentation:** 3 good
**Contribution:** 3 good
**Rating:** 8
**Confidence:** 3

**Summary:**

This paper proposes a new attention mechanism that can be realized to be linear in time. The proposed method uses vector quantization (mapping with learnable smooth codebooks) in a way that has non-zero gradients following [van den Oord *et al.* (2017)). Showing that attention as expressed in Definition 3.1. can be realized in linear time is not trivial, so the authors show in Thm. 3.7 how this speed up can be implemented using "cache" variables.

**Strengths:**

* Practicality: Instructions on how to implement the architecture is very detailed, and it looks like the authors experimented quite a lot to design an architecture that works well with various architecture sizes (190M, 1.2B, 1.3B parameters)
* Large-scale experiments across various challenging tasks.

**Weaknesses:**

The authors experiment with various architecture sizes; how did the authors choose which parameters to scale up? Is there some "scaling laws" that the authors observed to be useful? For example, fixing the codebook size to 256 may restrict the model's expressiveness which can be extremely critical for generative models. Given that the authors are emphasizing a decoder model which is particularly useful for generative tasks, I believe this paper would benefit from analysis on this new parameter's effect in addition to Sec. 5.1.1. One type of experiment that could demonstrate this effect would be to see if the model's performance saturates at some parameter count, and see if performance saturation is caused by the codebook size by training the same model with a larger codebook size S. Sharing these would be useful to anyone who plans to use this architecture for custom tasks.

Overall the paper is well written, but I think the paper could benefit if the authors provide intuition for the caching mechanisms and the implementation for efficient attention (Theorems 3.4-3.7). Further, the proofs are written just as a sequence of equations without any conceptual descriptions, so it's quite hard to understand what steps are being used in the proofs.

**Questions:**

* In Sec. 5.1.2 the authors say that the codebook size is 256 but Table 6 in the appendix shows S is 512. Is this a typo?
* What are the authors trying to show with ImageNet64? Is this part trying to show-case the model's applicability as a "vision transformer" type of model?

---

> ### Author Response · Authors · 2023-11-15
> **Reply to reviewer 1jb5**
>
> Thank you for your comments. We have recently uploaded a new version of our paper, which we believe is further improved; see our comment at the top for a summary of changes. We will now address your individual comments.
>
> > The authors experiment with various architecture sizes; how did the authors choose which parameters to scale up? Is there some "scaling laws" that the authors observed to be useful?
>
> In general, we tried copying hyperparameters from whatever paper was SoTA/near SoTA, then made adjustments specific to the design of GAU [1]. If the settings were slow or nonperformant, we tried other papers. Two examples that didn't work for us were the Focus Attention paper [2], which only used 39M parameters, and was slow to optimize even when the x-axis was wall clock time, and the 60-layer model for ImageNet64 used by Perceiver-AR [3], which was slower per step than a wide but shallower model, and was also slower to optimize when the x-axis was wall clock time.
>
> For our small model, we ultimately adapted from Sukhbaatar et al., 2021 [4], but copied it as if the MLP fan-out factor was 4x, whereas their setting was larger. For our large model, we ultimately used a model adapted from Hutchins et al., 2022 [5], who train a PG-19 model with 24 layers and 1.3B parameters. From this, we inferred the model width $D_{m}$ was 2048. When applying GAU, one replaces the vanilla transformer's attention and MLP sublayers with one GAU each, both using value and gate widths $D_{v} = 2D_{m}$, so we simply set $D_{v} = 4096$ in our large model. Moreover, GAU always uses query/key width $D_{k} = 128$, so we used this setting too. This strategy yields a model with roughly the same number of parameters, and in our experiments it worked quite well. The only difference between our 1.3B and 1.2B architectures are the token embeddings, since the PG-19 model has a vocabulary size of 32,000.
>
> For the codebook hyperparameters, we found EMA rate $\gamma = 0.99$ worked best, which is the same as the original VQ-VAE paper recommended. In addition, we found that using a small commitment loss coefficient $\beta = 0.0001$ worked reasonably well for all of our models, but we did not sweep over this choice when training our larger models.
>
> > ... One type of experiment that could demonstrate this effect would be to see if the model's performance saturates at some parameter count, and see if performance saturation is caused by the codebook size by training the same model with a larger codebook size S. Sharing these would be useful to anyone who plans to use this architecture for custom tasks.
>
> We agree it would be beneficial to be able to have a scaling law to predict the validation loss from the codebook size, non-codebook/non-embedding parameter count, and training token count. Ideally, this could be combined with a prediction of the time per training step, and then we could determine the optimal codebook size yielding the lowest loss under a given wall-time budget for training.
>
> At present, we feel do not have enough data to claim we have a formal scaling law. However, based on our ablations so far, it does appear that the logarithm of validation loss decreases linearly with the logarithm of the codebook size. In other words, the validation loss w.r.t. codebook size appears to follow a power law, $L(S) = (\frac{S_{0}}{S})^{\alpha}$. Similar to Kaplan et al., 2020 [6] we know this model is misspecified, since it implies a zero loss for an infinite codebook size. Adding a constant term similar to Henighan et al., 2020 [7] and Hoffman et al., 2022 [8] solves this problem, and it implies improvements in the logarithm of the validation loss will eventually saturate.
>
> That being said, we don't want to fit a log-linear model on the basis of 3 datapoints, especially when they're derived from a small dataset like Enwik8 where overfitting could lead to confounded results. To perform a proper study of this matter, it will be necessary to use larger datasets and many additional training runs. For this reason, we did not add a formal scaling law in our paper.
>
> >  the paper could benefit if the authors provide intuition
>
> The original idea was that the exact attention weights may not matter a lot, and they can be quantized as scalars. However, compressing the bit width of a float32 attention weight scalar can save at most a 32x constant factor on the attention compute using e.g., stochastic rounding when computing QK^T [9]. In contrast, using vector quantization and compressing the "duplicate" keys allows a linear savings, increasing with sequence length, and has zero variance.
>
> [1] https://arxiv.org/abs/2202.10447
>
> [2] https://arxiv.org/abs/2305.14952
>
> [3] https://arxiv.org/abs/2202.07765
>
> [4] https://arxiv.org/abs/2105.06548
>
> [5] https://arxiv.org/abs/2203.07852
>
> [6] https://arxiv.org/abs/2001.08361
>
> [7] https://arxiv.org/abs/2010.14701
>
> [8] https://arxiv.org/abs/2203.15556
>
> [9] https://eprints.maths.manchester.ac.uk/2790/

---

> ### Author Response · Authors · 2023-11-15
> **Reply to reviewer 1jb5, continued.**
>
> We hope our comment above helps shed some light on how VQ-Attention was derived initially, and how it ultimately works. It first applies vector-quantization to map similar keys to the same elements in a finite set of choices, then it deduplicates the keys while keeping the attention result the same as not deduplicating them.
>
> > Further, the proofs are written just as a sequence of equations without any conceptual descriptions, so it's quite hard to understand what steps are being used in the proofs.
>
> We will update the paper to be a bit clearer and will comment again to let you know when we have done so.
>
> > In Sec. 5.1.2 the authors say that the codebook size is 256 but Table 6 in the appendix shows S is 512. Is this a typo?
>
> No it's not; at the beginning of Section 5 (which was on the previous page) we wrote that the hyperparameters are given by Appendix B "unless specifically noted". Since we noted the choice of $S = 256$, basically we didn't feel the need to make any additional remark about this difference from our main hyperparameters.
>
> In Sec 5.1.1 and 5.1.2 we are trying to see the cumulative impact of the choices that weaken the attention mechanism. We start with $S = 1024$, then decrease it by 2x twice, then we remove the compressive cache. If you want, we can also run an ablation removing the compressive cache from the $S = 512$ model. If you do want us to add this ablation, please let us know as soon as you can.
>
> > What are the authors trying to show with ImageNet64? Is this part trying to show-case the model's applicability as a "vision transformer" type of model?
>
> We were trying to see if the model can be used as a generative model on domains other than text. In text, the absolute position of distant tokens tends not to matter a lot, since for example localized re-phrasings can shift the absolute index of subsequent tokens without changing the high-level meaning. In contrast, in images things like the row index and column index matter a lot, and it was not clear to us how well Transformer-VQ would work, since its keys in any given layer are compressed to a smaller set of options than the length of the sequence.
>
> Our main source of optimism for training on ImageNet64 was Perceiver AR, which also compresses the tokens into a relatively small representation space. Of course, Perceiver AR still takes quadratic time to generate a sequence due to the repeated applications of cross attention from each chunk.
>
> As an aside, since the parameter count of our ImageNet64 model is not known in comparison to Perceiver AR, we are also training a small model on this dataset right now, with a similar parameter count as Efficient VDVAE. Since this will not impact our main results, we assume it is okay to add this result to the paper.

---

> ### Author Response · Authors · 2023-11-17
> **Update: added clearer proofs and additional explanations**
>
> Since you mentioned the original proofs were terse, we went through all of them, added commentary, and generally made them easier to follow. Apologies for the delay, and hope you find the updated proofs informative.
>
> We have also made many more improvements to the latest manuscript, which are catalogued in our comment at the top of the page.
>
> If there's anything else we can do to help, please let us know!

---

> ### Author Response · Authors · 2023-11-21
> **Updates on the intuition, diagrams, and pseudocode**
>
> We have included two diagrams now (Figures 1 and 2) and have added pseudocode for our model in Appendix E.
>
> In addition to the improved readability of our proofs, we hope you find these additions improve the intuitiveness of our results.
> Thank you for your feedback, and please let us know if there's anything else we can do to help!

---

> > ### Comment · Reviewer_1jb5 · 2023-11-22
> >
> > Thanks for the response. I keep my rating, but I encourage the authors to improve how the intuition is described. I think it would be better to create new figures in place of updated figures 1 & 2, for example, as a graphical illustration of just the main components written in Theorem 3.7 to explain the caching variables and how VQ-attention results in linear time complexity. I don't understand what Figure 1 is now trying to do - illustrate vector quantization? Same for Figure 2: I think it would be better to illustrate why the identity holds, instead of dedicating space to show the shapes of the associated quantities.

---

> ### Author Response · Authors · 2023-11-22
> **Clarification on the diagrams**
>
> Hello and thanks for the reply.
>
> For your reference, we summarize the intention of each figure below, then discuss possible improvements.
>
> - In Figure 1, the colorful boxes were intended to represent hypothetical 3-dimensional keys, with their RGB intensities representing individual dimensions in the standard basis. The empty boxes were intended to represent the attention weights in response to a hypothetical query (not depicted). VQ-Attention distorts the attention weights, notably causing the weights for two initially-distinct "green" vectors, $k_{2}$ and $k_{5}$, to match after quantization. A consequence of these matchings is there are fewer attention scores to be computed (in general, $TS$ instead of $T^{2}$).
>
> - Figure 2 tries to show where the speedup comes from on a linear-algebraic level. VQ-Attention computes a small matrix product equivalent to a bigger one. It works because when using VQ on the keys, the attention matrix $\phi_{w}(\mathbf{Q}\hat{\mathbf{K}}^{\top})$ contains only $S$ unique column vectors when $\phi_{w}$ is an elementwise nonlinearity like $\exp(\cdot)$. As a result, the attention matrix can be obtained by computing the unique columns $\phi_{w}(\mathbf{Q}\mathbf{C}^{\top})$, then right-multiplying by $\boldsymbol{\Delta}$ to copy them to the correct columns of $\phi_{w}(\mathbf{Q}\hat{\mathbf{K}}^{\top})$. For efficiency, we use the associativity of matrix products to instead left-multiply $\mathbf{V}$ by $\boldsymbol{\Delta}$. Figure 2 was inspired by the Performer paper [1], though their diagram additionally had time complexities attached.
>
> As for the cache variables, the numerator cache $\mathbf{U}$ used in Theorems 3.6-3.7 is essentially a "partial" matrix product version of $\boldsymbol{\Delta}\mathbf{V}$ as required for causality (the sum over the reduced axis is a partial sum). The denominator cache $\mathbf{L}$ in Theorem 3.7 is essentially the same thing with a ones matrix in place of $\mathbf{V}$; it  contains counts of the codes for timesteps seen by the numerator cache so far, and is used to ensure the denominator of our softmax-based attention is correct.
>
> We are considering a few ideas to improve the visualizations, and will endeavor to update the final paper with the option deemed best:
> - For Figure 1, we will attempt to improve the caption.
> - For Figure 2, we will investigate making the color scheme more informative, e.g., by making the columns of $\phi_{w}(\mathbf{Q}\hat{\mathbf{K}}^{\top})$ varying colors, with duplicates.
> - Adding a diagram with a concrete example.
> - Adding a diagram showing the "column duplication" action by $\boldsymbol{\Delta}$ when computing $\phi_{w}(\mathbf{Q}\{\mathbf{C}}^{\top}) \boldsymbol{\Delta} = \phi_{w}(\mathbf{Q}\hat{\mathbf{K}}^{\top})$.
> - Adding a diagram showing the full lifecycle of the cache for a single iteration (one block).
>
> Thank you again for your feedback!
>
> References:
>
> [1] https://arxiv.org/abs/2009.14794

---

### Official Review · Reviewer_5aiB · 2023-10-31

**Soundness:** 2 fair
**Presentation:** 2 fair
**Contribution:** 2 fair
**Rating:** 6
**Confidence:** 2

**Summary:**

* The paper proposes a linear-time attention mechanism based on vector-quantization.
* This scheme is evaluated quantitatively and qualitatively on various datasets.

**Strengths:**

* The idea of using vector-quantization to achieve linear attention sounds reasonable.
* The paper includes a detailed discussion of various related works and how they differ from Transformer-VQ.
* Code is provided, aiding reproducability.
* The paper provides qualitative samples of the various trained models.

**Weaknesses:**

* I think the presentation of the paper could be improved. I think some pseudocode or diagrams illustrating the main ideas would be useful, while some of the theorems could potentially be moved to the Appendix.
* It seems to me that Transformer-VQ only significantly outperforms prior work on ImageNet64 where it uses a 7x larger model than the second best model.
* It also not entirely clear to me what the real-world advantage of Transformer-VQ is, I assume that the attention mechanism is significantly faster for longer context due to being linear. If so, it would be useful to demonstrate this with some actual speedup experiments.
* The datasets Transformer-VQ is evaluated on are also not current mainstream LLM datasets (which is the main application of Transformers and attention at the moment); I think results for e.g. standard token-level language modeling on C4 or The Pile would be more convincing.

Overall, while I find the general idea reasonable, I am not fully convinced of it's effectiveness by the current set of experiments, hence I am leaning more towards rejection.

**Questions:**

See weaknesses.

---

> ### Author Response · Authors · 2023-11-15
> **Reply to reviewer 5aiB**
>
> Thanks for your comments. We have recently uploaded a new version of our paper, which we believe is greatly improved; see our comment at the top for a summary of changes. We will now address your individual comments.
>
> > I think some pseudocode or diagrams illustrating the main ideas would be useful
>
> We have added some pseudocode to Appendix E, please let us know if this helps clarify things. In addition, we will attempt to create a satisfactory diagram illustrating the main idea.
>
> > Transformer-VQ only significantly outperforms prior work on ImageNet64 where it uses a 7x larger model than the second best model.
>
> On Enwik8, Transformer-VQ reaches parity with Transformer-XL using 33% fewer parameters and a 75% shorter cache.
>
> On PG-19, Transformer-VQ nearly matches the performance of Block-Recurrent Transformers while using a vocabulary with a higher coverage rate of rare characters in the training set (100% vs 99.95% [1]), and Adafactor with a standard zero coefficient on first moment estimate. In contrast, Block-Recurrent Transformers uses an expensive and nonstandard nonzero coefficient on the unfactored first moment [2]. We only discovered these differences after our costly training run had finished and we had evaluated on the test set, so our PG-19 model might actually be better if we had trained under comparable conditions. Finally, Transformer-VQ is amenable to a speedup using the method of Hua et al., 2022 [3] for intra-block sums and cross-block reductions, while Block-Recurrent Transformers use a general scan and are not mathematically amenable to such a speedup technique.
>
> On ImageNet64, Transformer-VQ uses a large model because our goal was to match the perceptual quality of Perceiver-AR [4] which trains on the old version of this dataset and did not disclose their parameter count, but presumably uses all of TPU memory. We trained for 33% fewer steps than Perceiver-AR, however, because we did not have time to conduct a learning rate sweep for the longer training schedule (we just reused our settings from PG-19). We are training a small model now, and are finding it to outperform the previous SoTA on the new ImageNet64 (Efficient VDVAE, [5]) while having roughly the same number of parameters. We plan to update the paper to include its final performance (it will be done in a few days); since it does not affect our main results, we assume this is allowed.
>
> > It also not entirely clear to me what the real-world advantage of Transformer-VQ is
>
> The advantage of Transformer-VQ is that it is a potential drop-in replacement for full attention, but is efficient to train on long sequences.
>
> > I assume that the attention mechanism is significantly faster for longer context due to being linear. If so, it would be useful to demonstrate this with some actual speedup experiments.
>
> We have added these experiments in our updated paper. Transformer-VQ runs 3x faster at sequence length 8192 with 190M parameters, runs 12x faster at length 32768, and can scale to length 131072 while maintaining similar throughput and without running out of memory.
>
> > The datasets Transformer-VQ is evaluated on are also not current mainstream LLM datasets (which is the main application of Transformers and attention at the moment); I think results for e.g. standard token-level language modeling on C4 or The Pile would be more convincing.
>
> We will address this comment in a separate reply due to space constraints.
>
> References:
>
> [1] To verify, download https://github.com/google-research/meliad/blob/main/transformer/vocabs/pg19train_bpe_32000.model
> and run https://gist.github.com/transformer-vq/ea2d1d07c556c6862b2b95122de60511
>
> [2] https://github.com/google-research/meliad/blob/main/optimizer_config.py#L70
>
> [3] https://arxiv.org/abs/2202.10447
>
> [4] https://arxiv.org/abs/2202.07765
>
> [5] https://arxiv.org/abs/2203.13751
>
> [6] https://arxiv.org/abs/1904.10509

---

> ### Author Response · Authors · 2023-11-15
> **Reply to reviewer 5aiB, continued.**
>
> > The datasets Transformer-VQ is evaluated on are also not current mainstream LLM datasets (which is the main application of Transformers and attention at the moment); I think results for e.g. standard token-level language modeling on C4 or The Pile would be more convincing.
>
> The PG-19 and ImageNet64 datasets we evaluate on are standard long-context modeling benchmarks and are in fact the largest publicly-available autoregressive modeling datasets used in well-cited and spotlighted long-context papers from 2022 [3][4][7][8].
>
> As for the Pile, it is a great dataset for training LLMs, yet there are several reasons we didn't use it:
> 1. At 800GB, it requires significant engineering overhead to download and monetary cost to store.
> 2. The download link has been broken for some time [9].
> 3. The average document length is only 5.9 KiB, or 6000 bytes [10], which is at most a few thousand BPE tokens.
> 4. Only one of the efficient sequence modeling papers we cited used The Pile, and did not report any perplexity improvement beyond 2^12 (4096) tokens, despite training with an 8k context length [11].
>
> As for C4, it also has a size of 800GB, it is mainly used for masked language modeling, and its document lengths again tend to be relatively short. We are only aware of one paper credibly using C4 for long-context autoregressive modeling [8], and they used a filtered version of it with 4k tokens as the minimum document length. Since 4k is still quite short, this dataset is not very interesting for long-context modeling, and as far as we know it has not been used since.
>
> As we experienced minimal issues with overfitting on PG-19 and ImageNet64, which are more popular benchmarks for long-context autoregressive modeling, we do not believe that evaluating on either of these near-terabyte-scale datasets would yield any particular insights at our current scale, especially given their short average document lengths.
>
> Lastly, please note that PG-19 is a BPE subword-token language modeling task and involves 11GB of text. It is only the perplexity that is converted to word-level for benchmarking purposes, per Rae et al., 2020 [12].
>
> References:
>
> [3] https://arxiv.org/abs/2202.10447
>
> [4] https://arxiv.org/abs/2202.07765
>
> [7] https://arxiv.org/abs/2203.07852
>
> [8] https://arxiv.org/abs/2203.08913
>
> [9] https://the-eye.eu/public/AI/pile/
>
> [10] https://github.com/EleutherAI/the-pile
>
> [11] See Figure 5 of https://arxiv.org/pdf/2305.13048.pdf
>
> [12] https://arxiv.org/abs/1911.05507

---

> ### Author Response · Authors · 2023-11-16
> **Update: added the diagram**
>
> In addition to providing the requested pseudocode, we have updated the paper with a schematic illustration of the VQ-Attention approximation, and captioned it in a way that explains it intuitively.

---

> ### Author Response · Authors · 2023-11-17
> **Updates: added ImageNet64 with small model, additional speed tests, additional pseudocode**
>
> We are writing to let you know that we have added ImageNet64 results for a small model. Obtaining 3.22 BPB, it significantly outperforms the previous SoTA result by Efficient VDVAE, despite being nearly the same size!
>
> In addition, we have added additional speed tests for various methods of computing cache variables. The fastest method also happens to be the one that's linear time.
>
> Lastly, we have added pseudocode for each of the various methods for computing the cache variables, and clarified some additional ambiguities about how we keep our method numerically stable.
>
> Thanks for taking the time to review, and please let us know if there's anything else we can help with!

---

> ### Author Response · Authors · 2023-11-21
> **Added second diagram**
>
> In addition to the requested speed tests (Appendix B), pseudocode (Appendix E), and initial diagram (Figure 1), we have included additional diagram (Figure 2) to show how our method reduces the computational cost of attention on long sequences. We hope you find these changes to improve the presentation of our paper.
>
> Thanks for your feedback, and please do let us know if there's anything else we can do to help!

---

> > ### Comment · Reviewer_5aiB · 2023-11-22
> > **Reviewer Response**
> >
> > Thank you for the detailed replies!
> >
> > I appreciate the effort spent on improving the paper and think the diagrams, pseudocode, proof reorganization and speedup results have significantly upgraded the presentation.
> >
> > However, I am still not fully convinced by the experiments. It seems to me that, as further explained by the authors, many results are being compared even though they are not directly comparable for various technical reasons. While I understand some of the challenges that the authors faced here, I think setting up fully controlled experiments (same model size, same training budget, same architecture swapping only the new attention mechanism, same loss function, etc.) against a few of the strongest baselines would have been preferable. Also, the strongest baseline in Table 5 appears to be some convolution-based hierarchical auto-encoder while Transformer-VQ is an autoregressive Transformer, making it a bit hard to tell where exactly the improvements are coming from.
> >
> > As for the C4/Pile datasets, I would have been primarily interested in whether Transformer-VQ is able to keep up with optimized baselines in a large-scale standard range setting, where basic Transformers are very strong. However, an architecture optimized particularly for very long-range and smaller datasets is also a reasonable, albeit less impactful, contribution; it was initially not clear to me that this is the focus of the work.
> >
> > Overall, I will increase my score slightly and encourage the authors to provide perhaps an additional Table in the Appendix with information on the various baselines (size, training steps, architecture, etc.) to provide a better picture to the reader what exactly is being compared here.

---

> ### Author Response · Authors · 2023-11-23
> **Follow-up reply**
>
> Thank you, we appreciate your increase in score!
>
> > It seems to me that, as further explained by the authors, many results are being compared even though they are not directly comparable for various technical reasons.
>
> Our Enwik8 and PG-19 results are fully comparable. It just so happens we found overfitting to be a challenge on Enwik8 and used a tougher choice of optimizer/vocabulary on PG-19. This does not in any way diminish our results; it strengthens them.
>
> On ImageNet64, we sought to use models with similar size to Perceiver AR and Efficient VDVAE, and we train for fewer steps than either of them: they use 750k and 1.6m steps, while we used large and small models trained for 500k and 125k steps.
> The only issue is that Perceiver AR uses a different version of the dataset, now unavailable via official channels. However, given our extremely persuasive results versus Efficient VDVAE, we feel our results are already quite strong.
>
> > same model size, same training budget, same architecture swapping only the new attention mechanism, same loss function, etc.
>
> Given the diversity of the baselines for each dataset, a faithful and fair implementation of all the baselines would have slowed down our project significantly, and required much more compute to obtain results for. It also would have taken considerable development overhead, and would have greatly complicated our configuration pipeline. There really wasn't a great option here, so we chose to defer to the results reported by each paper.
>
> As far as training with "the same loss function" goes, this may be taking it a bit too far: our model needs a commitment loss, and some other baselines use their own idiosyncratic auxiliary losses, such as the z-loss in Perceiver AR and the sparsity loss in Expire-Span.
>
> > a large-scale standard range setting
>
> We assume this is a reference to LLM pretraining on short-context documents. We indeed focus on training from scratch on long contexts, but in future work may investigate the paradigm of short-context pretraining and long-context finetuning.
>
> > encourage the authors to provide perhaps an additional Table in the Appendix with information on the various baselines (size, training steps, architecture, etc.)
>
> Certainly, we would be happy to include a table with this information in the final version.
>
> Thanks again for your feedback!

---

### Author Response · Authors · 2023-11-15
**Updates to paper**

Change log for revisions:

Version 2 (November 15, 2023 at 5pm UTC)
- Included training latency and throughput benchmarks versus quadratic-time attention. Benchmarks evaluate single-headed gated attention, multi-query attention, and multi-head attention for Transformer-VQ and the vanilla transformer. Sequence lengths range from 2048 to 131072.
- Moved qualitative analysis to the Appendix D.
- Added pseudocode for VQ-Attention in Appendix E.
- Added placeholder for final result of 190M ImageNet64 model. Training for this model is progress, uses the existing hyperparameters, and the model has already reached parity with previous SoTA, Efficient VDVAE. (Training will conclude in a few days.)
- Released optimized version of Transformer-VQ at https://github.com/transformer-vq/transformer_vq/tree/optimized
- The legacy version of Transformer-VQ uses jax.lax.scan similar to Block-Recurrent Transformers (Hutchins et al., 2022), while the optimized version uses intra-block sums and cross-block reductions similar to FLASH (Hua et al., 2022). Both versions outperform their corresponding versions of quadratic-time full attention; the paper includes results for the optimized version of both attentions because using a scan for quadratic-time attention is unnecessary.
- Misc: backpropagation window name is now W; abbreviation of GAU is corrected to gated attention unit; additional references on kernelized attention added to the related work section; discussion of post-training quantization expanded in the related work section; conclusion shortened due to space constraints.

Version 3 (November 16, 2023 at 6am UTC)
- Added schematic illustration of the VQ-Attention approximation.
- Tidied up the experiment section, since the descriptions and analysis were taking up too much space.

Version 4 (November 17, 2023 at 5pm UTC)
- ImageNet64 result for small model has been added: sets a new SoTA of 3.22 bpb for small models.
- Proofs are now easy to read and understand.
- Added pseudocode for each cross-block reduction method.
- Added complete throughput benchmarks for Full Attention vs VQ Attention.
    For VQ-Attention, we evaluated cross-block reductions for cache variables based on serial scan, matmul, and associative scan.
    For both VQ and Full Attention, we evaluated single-head gated attention, multi-query attention, and multi-head attention.
    Speedup of using VQ-Attention vs Full Attention was >3x at length 8192, and >12x at length 32768.
    VQ-Attention even maintains similar throughput at sequence length 131072.
- Added additional throughput benchmarks using input-scanning.
- Added additional remark to the abstract.
- Added additional references to state-space models in the introduction.
- Illustration in the introduction is respaced; caption now provides a bit of intuition about why VQ-Attention actually works, not just how.
- Cleaned up the appendix spacing.

Version 5 (November 20, 2023 at 4pm UTC)
- Clearer explanation of straight-through estimator.
- Misc typos in corrected.
- Improved discussion of scaling laws following the codebook size ablations.
- Expanded conclusion with directions for future work.
- Additional diagram and model-parallel codebase are still a work in progress!

Version 6 (November 21, 2023 at 5am UTC)
- Added additional diagram showing how VQ-Attention reduces the attention computations to linear time.
- Improved ImageNet64 analysis and conclusion.

Version 7 (November 22, 2023 at 6am UTC)
- Improved figure captions for the two diagrams.
- Improved intro, conclusion.

---

### Meta-Review · Area_Chair_DmEU · 2023-12-06

**Metareview:**

Paper propose finite codebook-based Vector-Quantization for attention mechanism  key vectors. This theoretically reduce the attention complexity from quadratic to linear. This leads and efficient caching leads 3-12x faster models with respectable performance. However, authors are highly encouraged to improve their presentation and add additional details and experiments as suggested by the reviewers.

**Justification For Why Not Higher Score:**

Results and idea presentation can be improved.

**Justification For Why Not Lower Score:**

Potentially useful result for scaling up transformers

---

### Decision · Program_Chairs · 2024-01-16

Accept (poster)